# A bimetallic nanoplatform for STING activation and CRISPR/Cas mediated depletion of the methionine transporter in cancer cells restores anti-tumor immune responses

Ying Huang[1,2], Geng Qin[1,2] ✉, TingTing Cui[1,2], Chuanqi Zhao[1,2] ✉, Jinsong Ren[1,2] & Xiaogang Qu ●[1,2] ✉

Lack of sufficient cytotoxic T lymphocytes (CD8[+] T cells) infiltration and dysfunctional state of CD8[+] T cells are considered enormous obstacles to anti-tumor immunity. Herein, we construct a synergistic nanoplatform to promote CD8[+] T cell infiltration in tumors while restoring T cell function by regulating methionine metabolism and activating the STING innate immune pathway. The CRISPR/Cas9 system down-regulates the methionine transporter SLC43A2 and restricts the methionine uptake by tumor cells, thereby relieving the methionine competition pressure of T cells; simultaneously, the released nutrition metal ions activate the cGAS/STING pathway. In this work, the described nanoplatform can enhance the effect of immunotherapy in pre-clinical cancer models in female mice, enhancing STING pathway mediated immunity and facilitating the development of amino acid metabolic intervention-based cancer therapy.

Cancer immunotherapy has revolutionized cancer treatment[1–3]. Despite remarkable achievements accompanied by therapeutic efficacy, patients have experienced limited response to immunotherapies[4]. CD8[+] T cells, as the main immune cell type, can eradicate cells that have undergone malignant transformation[5–7]. However, in an immunosuppressive tumor microenvironment, CD8[+] T cell infiltration is insufficient and tumor-infiltrating CD8[+] T cells are often dysfunctional[8,9]. This has motivated the need for strategies to increase CD8[+] T cell infiltration while restoring CD8[+] T cell function in tumors.

Metabolic pressure on tumor-infiltrating CD8[+] T cells is a critical cause of CD8[+] T cells dysfunction and loss of potential antitumor activity[10,11]. As key nutrients for T cells, amino acids play important roles in protein synthesis and numerous metabolic pathways[12–14].

Recently, Bian et al. explored the link between abnormal amino acid metabolism and CD8[+] T cell dysfunction[15]: Among various amino acids, methionine deficiency caused the most significant CD8[+] T cell death and dysfunction[16]. This means that the acquisition of methionine is crucial for the survival and function of CD8[+] T cells. However, tumor cells compete with CD8[+] T cells for methionine through fanatical methionine uptake to impair CD8[+] T cell survival and function. Unlike CD8[+] T cells that rely primarily on SLC7A5 for methionine transport, tumor cells uptake methionine fanatically through high-expressing SLC43A2[15,17]. However, there is no effective inhibitor that can target SLC43A2 currently. As a powerful genome-editing tool, the CRISPR/Cas9 provides an opportunity to inhibit SLC43A2 expression[18–26]. CRISPR/Cas9 can downregulated SLC43A2 expression to normalize

[1]Laboratory of Chemical Biology and State Key Laboratory of Rare Earth Resource Utilization, Changchun Institute of Applied Chemistry, Chinese Academy of Sciences, Changchun, Jilin 130022, P. R. China. [2]School of Applied Chemistry and Engineering, University of Science and Technology of China, Hefei, Anhui 230026, P. R. China. ✉e-mail: qingengjl@sohu.com; chuanqizhao12@yahoo.com; xqu@ciac.ac.cn

CD8[+] T cell methionine metabolism. Compared with conventional methods such as siRNA or shRNA, the CRISPR/cas9 system has different advantages. CRISPR/Cas9 has the advantages of permanently modifying target genes, being exceedingly selective, and having low off-target likelihood. It is one of the most efficient, simple, and low-cost gene editing technologies available, and is a very popular gene editing system present. This approach can deprive tumor cells of methionine uptake, relieve the methionine metabolic presses on CD8[+] T cells, and restore the function of tumor-infiltrating CD8[+] T cells. Additionally, depriving tumor cells of methionine supply can also synergize with multiple therapeutic modalities by disrupting nucleotide metabolism and redox balance[27–30].

Many studies have demonstrated that activation of innate immunity is crucial for the priming of antigen-specific T cells and subsequent T cell infiltration[31–34]. As one of the primary innate immune pathways, the cGAS/STING pathway attracted extensive attention due to its bridging role between innate and adaptive immunity[35,36]. The cGAS/STING pathway can promote the secretion of type I-interferon (IFN) and the production of pro-inflammatory cytokines to increase tumor infiltration of antitumor T cells. Since cytosolic DNA from damaged DNA molecules could be recognized by cGAS to prime the STING pathway[37], a variety of therapies that increase the release of damaged DNA fragments have been used in combination to enhance the priming of the STING pathway, such as radiotherapy[38,39], photodynamic therapy (PDT)[40,41], chemotherapeutic[42], radiofrequency ablation (RFA)[43] and so on[44–46]. Methionine restriction can enhance the sensitivity of tumor cells to oxidative stress, so we speculate that methionine restriction would increase the release of damaged DNA fragments in cells and contribute to activate the STING signaling pathway.

Metal-organic frameworks (MOFs) refer to porous materials constructed by bridging metal ions or metal clusters with organic ligands[47,48]. MOFs can also provide nanocarriers for nucleic acid therapy[49,50]. Coincidentally, metal ions detached from biodegradable MOFs could also be involved in immune processes. Nutritional metal ions are crucial in many important immune processes[51]. For instance, manganese ions ($Mn^{2+}$) and zinc ions ($Zn^{2+}$) could greatly enhance the cGAS/STING signals. (Free $Zn^{2+}$ can promote the enhancement of cGAS enzymatic activity[52]; $Mn^{2+}$ can not only enhance the sensitivity of cGAS to double-stranded DNA (dsDNA), but also enhance STING activity by enhancing cGAMP-STING binding affinity[53].) In addition, transition metal ions can also induce reactive oxygen species (ROS) storms. ($Zn^{2+}$ can enhance the generation of $\cdot O^{2-}$ and $H_2O_2$ by inhibiting the mitochondrial electron transport chain, achieving the rapid accumulation of endogenous ROS[54,55]; $Mn^{2+}$ can transform the accumulated endogenous $H_2O_2$ into the strong toxic $\cdot OH$[56,57].) ROS storm would damage mitochondria and releases mitochondrial DNA (mtDNA), thereby promoting the activation of the STING signaling pathway[58,59]. MOFs inheriting metalloimmunology exhibited great potential in cancer immunotherapy.

Here, we show a nanoplatform, which could reverse the insufficient and dysfunctional tumor-infiltrating T cells, established by encapsulating a CRISPR plasmid to downregulate the methionine transporter into Mn/Zn bimetallic MOF nanoparticles. In this design (Fig. 1), the CRISPR plasmid enhances activation of the cGAS/STING pathway and reverses exhaustion of T cells, by affecting methionine metabolism in tumor cells. Simultaneously, the nutrient metals released by MOF (Mn/Zn-ZIF-8) not only induce ROS storm to kill tumors, but also stimulate the activation of the STING pathway. Hyaluronic acid (HA) modification is used to enhance tumor recognition and accumulation. The bimetallic nanoplatform can not only realize the effective delivery of CRISPR/CAS9 plasmids but also greatly optimize immunotherapy by activating the cGAS/STING signaling pathway and targeting methionine metabolism.

## Results

### PMZH preparation and characterization

First, we constructed a CRISPR/CAS9 plasmid (pDNA) with the function of gene-editing methionine transporter by cloning sgRNA targeting the *SLC43A2* into plasmid PX330 (Fig. 2a). Mixing pDNA during the process of mixing metal ions and organic ligands to form nanoparticles can encapsulate the pDNA in the whole nanoparticle instead of the unstable surface adsorption, thus making the pDNA encapsulation more firmly. The $Mn^{2+}$–4 N tetrahedral geometry is unstable, which leads to some difficulties in directly incorporating $Mn^{2+}$ into ZIF-8[60]. Therefore, we first mixed the pDNA, Zn $(NO_3)_2$ and 2-methylimidazole in an aqueous solution with stirring to synthesize pDNA@ZIF-8 (PZ) nanoparticles by the one-step method. Subsequently, manganese ions were introduced into the framework using a post-synthesis exchange method to form the pDNA@Mn/Zn-ZIF (PMZ) for further biomedical applications[61,62]. An agarose gel retardation assay was used to study the binding affinity of pDNA in PMZ (Fig. 2b). To obtain appropriate transfection efficiency while protecting pDNA from nuclease degradation, Mn/Zn-ZIF (MZ) was efficiently complexed with pDNA at a weight ratio of 45:1. From the scanning electron microscopy (SEM) images (Supplementary Fig. 1) and transmission electron microscope (TEM) pictures (Fig. 2c), PMZ exhibited a uniform nanohydrangea structure with an average diameter of ~250 nm. Inductively coupled plasma-optical emission spectrometry (ICP-OES) results demonstrated that Mn/Zn ratio of MZ nanoparticles was 1:10, which was similar to the XPS results (Supplementary Fig. 2). The element mapping showed the spatial distribution of Mn, Zn, P, N, C, and O elements in the PMZ (Fig. 2c). These, as well as the uniform distribution of manganese in the elemental mapping images (Fig. 2c), indicated the successful preparation of bimetallic MOFs. The uniform distribution of p elements indicated that pDNA was successful encapsulated into MZ. Compared to MZ, encapsulation of pDNA resulted in a slight increase in the hydrodynamic diameter of the nanoparticles and a decrease in the ζ potential of the nanoparticles (from −10.86 mV to −16.43 mV) (Supplementary Figs. 3b and 4a). As the X-ray diffraction (XRD) illustrated, the nanoparticles still had a good crystal form after the incorporation of manganese, and the encapsulation of pDNA had no obvious effect on its crystal form (Fig. 2d). The Fourier transform infrared (FTIR) spectra demonstrated the successful coating of HA (Supplementary Fig. 5b). We refer to PMZ@HA as PMZH, MZ@HA as MZH. Modification of HA hardly affected the morphology, but slightly increased the hydrodynamic diameter of PMZ (Supplementary Fig. 3a, b). ζ potential and DLS of PMZH in serum were assesed at different times to investigate the stability of PMZH nanoparticles in the physical environment. As shown in Supplementary Fig. 3c, the DLS/ζ of PMZH remained stable within 48 h without significant changes, indicating that PMZH nanoparticles were highly stable in serum media under physiological conditions. The synthesized PMZH began to degrade under acidic conditions, and the pDNA (Fig. 2e) and metal ions (Fig. 2f) were mostly completely released after incubation at pH = 5.5 for 24 h. Under the pH condition that simulates the normal cell microenvironment, pDNA was hardly released, and a very small amount of manganese and zinc ions were dissolved from the PMZH. This ensured that pDNA and metal ions would not be released prematurely and cause uncertain side effects before reaching the tumor site.

### Assessment of lysosomal escape and gene editing ability of PMZH

We evaluated the cellular uptake capacity of PMZH prior to in vitro therapeutic studies. Flow cytometry analysis showed that FITC-labeled PMZH could be efficiently internalized by 4T1 cells (Fig. 3c). The targeting effect of HA enhanced the enrichment of FITC-modified PMZH in the 4T1 cells (Supplementary Fig. 6). After endocytosis, efficient lyso/endosomal escape is critical for gene editing[63,64]. The cellular uptake and lysosomal escape of PMZH could be visualized visually

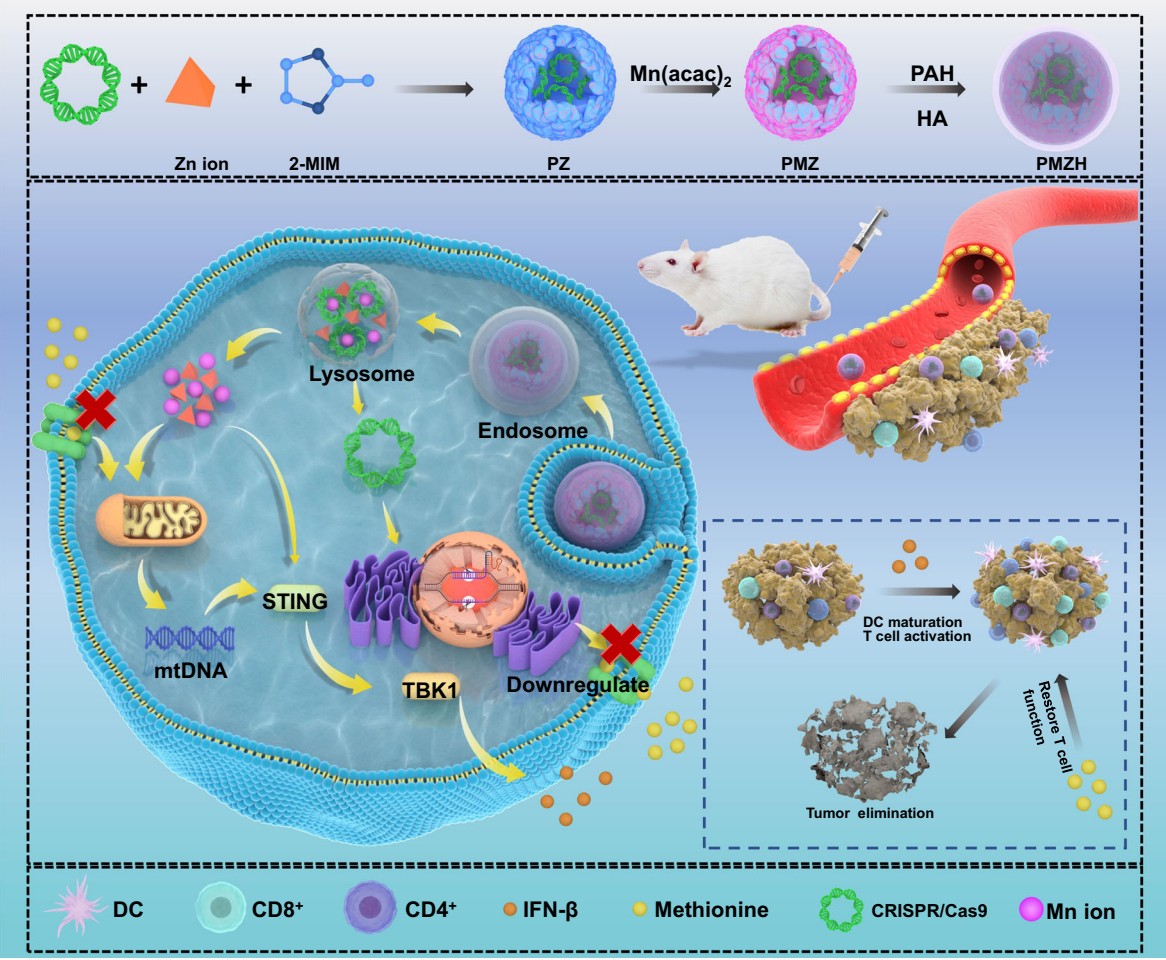

**Fig. 1 | Schematic illustration of PMZH nanoplatform formation and PMZH for methionine metabolism regulation, nutritional metal ion therapy, and immune stimulation.** 2-MIM 2-methylimidazole, DCs dendritic cell, PZ pDNA@ZIF-8, PMZ pDNA@Mn/Zn-ZIF, PAH poly (allylamine hydrochloride), HA hyaluronic acid, PMZH pDNA@Mn/Zn-ZIF@HA, mtDNA mitochondrial DNA, TBK1 TANK-binding kinase 1.

from confocal laser scanning microscopy (CLSM) images (Fig. 3a); The color scatter plots (Fig. 3a) and corresponding Pearson correlation coefficient (PCC) values (Fig. 3b) between red and green fluorescence signals further confirmed that PMZH achieved efficient lysosomal/endosomal escape. The imidazole groups in PMZH is the reason for its good endosomal/lysosomal escape ability. The proton sponge effect of this intracellular pH buffering element reduced lysosomal degradation, thus achieving superior endosomal/lysosomal escape[65–67]. The excellent endosome/lysosome escape property of PMZH provides a guarantee for the realization of CRISPR/Cas9-mediated genome editing.

Next, the transfection efficiency of PMZH in 4T1 cells was further investigated. The optimal dose was chosen by co-incubating different concentrations of MZH with 4T1 cells (Supplementary Fig. 7). When the concentration of MZH was lower than $120\,\mu g\,mL^{-1}$, the cell viability was greater than 80% after co-incubating with 4T1 cells for 12 h. Therefore, we chose to conduct subsequent cell transfection experiments at the concentration of $120\,\mu g\,mL^{-1}$. We selected the encoding enhanced green fluorescent protein (pEGFP) plasmid, which is similar in size to the CRISPR/Cas9 plasmid, as the reporter gene for evaluation. A fluorescence microscope (Fig. 3d, e) and flow cytometry (Fig. 3f) was used to analyze the transfection efficiency. The transfection efficiency of MZH (≈32%) was slightly lower than that of commercially available Lipofectamine 6000 (lipo)(≈50%), but still a good transfection efficiency. The good transfection efficiency gave confidence to further explore its gene editing efficiency. In addition, Sanger sequencing

analysis showed (Fig. 3g) that the PMZH-treated group displayed obvious mutational peaks at the target location, also confirming the mutation of the *SLC43A2* locus. Subsequently, we detected the expression of SLC43A2 protein in 4T1 cells after various treatments by immunoblot analysis. The results showed that PMZH could disrupt the protein expression of SLC43A2 in tumor cells efficiently via CRISPR/CAS9-mediated gene editing (Fig. 3h). We used pDNA inserting a non-targeting sgRNA as the control plasmid to construct CMZH. Neither gene expression nor protein expression of *SLC43A2* was affected by CMZH (Fig. 3g, h).

## PMZH restored T cell immunity through methionine metabolism regulation

The solute carrier family (SLC) can transport methionine into cells, among which the high expression of SLC43A2 transporter in tumor cells will transport methionine fanatically resulting in reduced methionine available to T cells[15]. However, methionine is a critical nutrient to T cells, and methionine deficiency would inevitably affect the survival and destroy the function of T cells[16,27,68]. We assessed the effect that competition for methionine by tumor cells confers on T cell viability (Supplementary Fig. 8). We first assessed the viability of T cells and 4T1 cells after 48 h of culture in media containing different concentrations of methionine (Supplementary Fig. 8a, b). As in previous reports, the acquisition of methionine was critical for T cell viability. Next, we co-cultured T cells and 4T1 cells in a Transwell system (Fig. 4b). As the methionine concentration in the culture medium

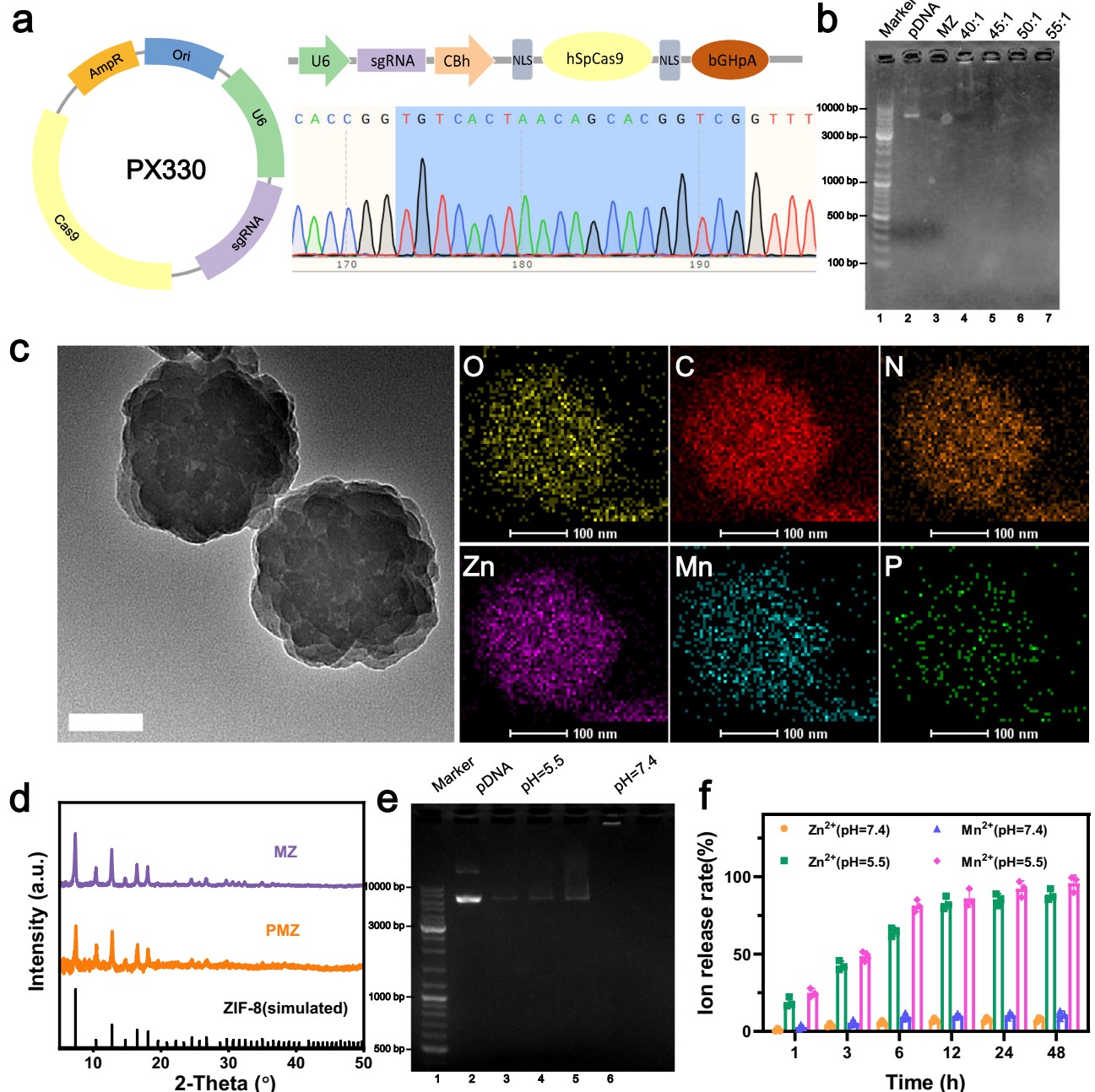

**Fig. 2 | Construction and characterization of PMZH. a** Schematic structure of PX330 vector after inserting sgRNA and sequencing result of the sgRNA targeting *SLC43A2* in CRISPR-Cas9 plasmid. **b** 1% agarose gel image. Lane 1-3, DNA ladder, naked pDNA, MZ, respectively; lane 4–7, MZ/pDNA at mass ratio of 40:1, 45:1, 50:1, and 55:1, respectively. **c** TEM image and corresponding element mappings of PMZ. Scale bar is 100 nm. **d** XRD analysis of PMZ/MZ nanoparticles. **e** Gel electrophoresis assay of PMZH after incubation in buffers with different pH. Lane 1-2, DNA ladder, naked pDNA, respectively; lane 3–5, PMZH incubation in acid buffers after 3 h, 6 h, 24 h, respectively; lane 6, PMZH incubation in buffers with pH 7.4 after 24 h. **f** Ion release at various time and under various conditions. $n = 3$ independent samples. Representative results showed in panel **b**–**e** were obtained from three independent samples. Data were presented as mean ± SD. Source data are provided as a Source Data file.

increased, 4T1 cell viability gradually recovered, but T cells did not (Supplementary Fig. 8c, d). This is due to the ability of tumor cells to compete for methionine in excess of T cells, thereby compromising T cell survival and function. Disruption of SLC43A2 in tumor cells by PMZH would affect methionine metabolism in T cells and tumor cells. We first assessed the ability of tumor cells to consume methionine after different treatments. The cell culture medium was collected and analyzed after incubation with different materials for 48 h. The methionine content in the culture medium was quantitatively analyzed by high performance liquid chromatography (Fig. 4a). We set up lipo,

P@lipo, pDNA, Mn²⁺, Zn²⁺, Mn²⁺/Zn²⁺, CMZH, and PMZH groups respectively to remove the effect of the antitumor ability of lipo and individual components of the PMZH on the methionine content in the medium. Compared with the lipo group and the CMZH group, both the P@lipo group and the PMZH group significantly reduced the consumption of methionine in the medium. Meanwhile, we also evaluated the effects of these groups on the viability of T cells alone and 4T1 cells alone (Supplementary Fig. 9). We then co-cultured T cells and differently treated 4T1 cells in the Transwell system. After 48 h of co-culture, the apoptosis rate was analyzed by flow cytometry (Fig. 4c, d), and the

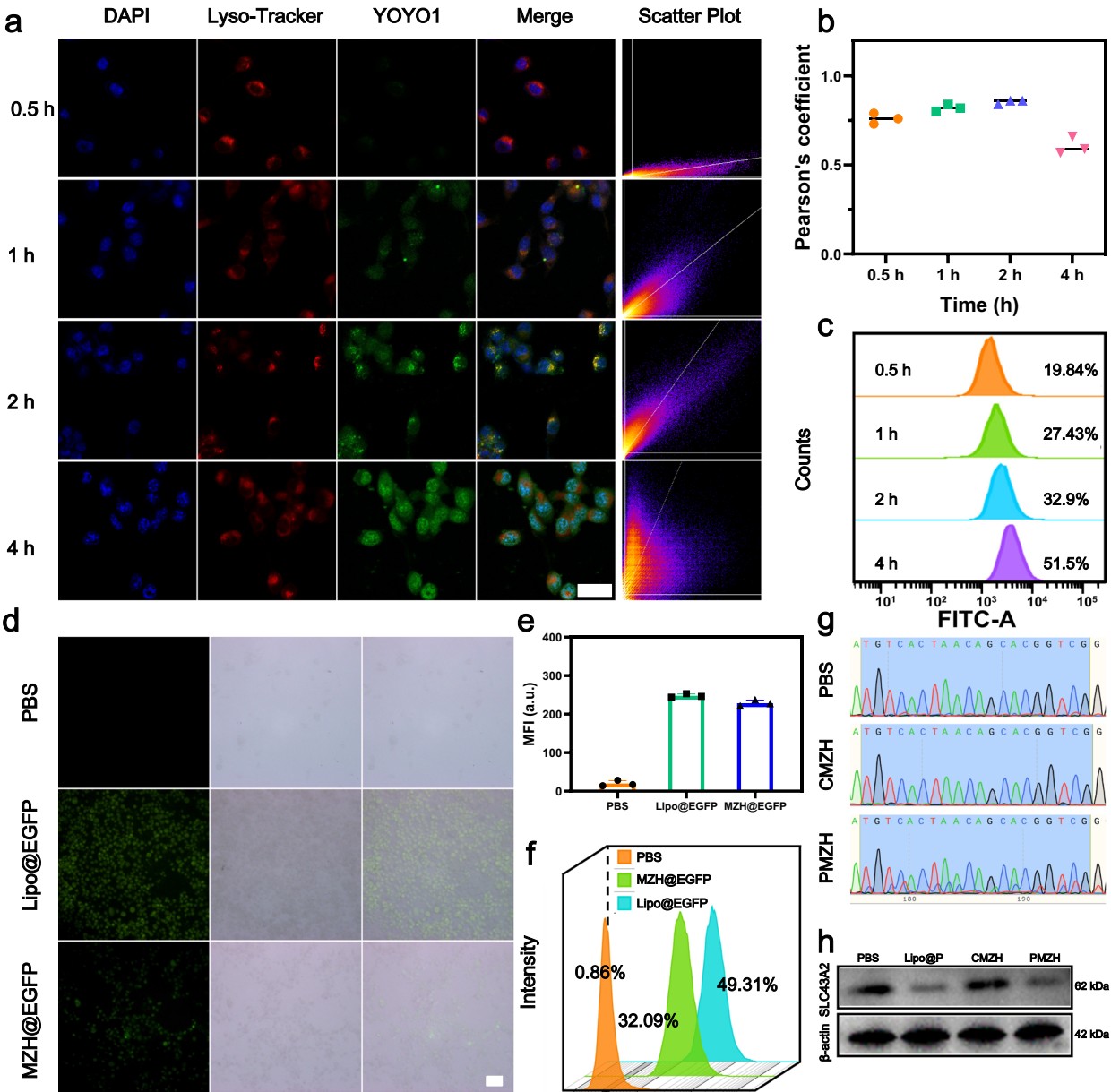

**Fig. 3 | Lyso/endosomal escape and gene editing. a** Confocal microscopy images of PMZH co-incubated with 4T1 cells for 0.5, 1, 2, and 4 h to assess lyso/endosomal escape capacity. Plasmids were labeled with YOYO-1 (green). The scatter plot is the relationship between fluorescence signals in red and green. **b** Pearson's correlation coefficient (PCC) values of PMZH colocalized with lysosomes. The PCC values were calculated using ImageJ software. **c** Assessing the cellular uptake capacity of FITC-PMZH using flow cytometry. **d** Evaluation and **e** Semi-quantitative of MZH@EGFP transfection efficiency by fluorescence microscopy. **f** Evaluation of MZH@EGFP transfection efficiency by flow cytometry. **g** Sanger sequencing of PCR amplicons of target loci after PBS, CMZH, PMZH treatment. **h** WB analysis of SLC43A2 expression after indicated treatments. (Scale bar of picture **a** and **d**: 50 μm). $n = 3$ independent experiments in **b** and **e**. Representative results showed in panel **a**, **c**, **d**, **f**, **h** were obtained from three independent samples. Data were presented as mean ± SD. Source data are provided as a Source Data file.

cytokines were analyzed by Elisa kit (Fig. 4e). The CMZH and PMZH groups exhibited a significant antitumor ability, which might be related to the ROS storm caused by $Zn^{2+}$ and $Mn^{2+}$. In the Transwell co-culture experiment, compared with the CMZH and lipo groups, P@lipo and PMZH groups had decreased T cell apoptosis and increased cytokines secretion. Whereas no difference was evident in T cell culture alone (Supplementary Fig. 9b). Since low methionine content decreases H3K79me2 in T cells, thereby impairing T cell immunity, we evaluated H3K79me2 content of T cells in the co-culture system after different treatments (Supplementary Fig. 10). Treatment of PMZH resulted in increased expression of H3K79me2 in T cells, suggesting that PMZH treatment contributed to the restoration of T cell

immunity. The excellent ability of PMZH to antitumor and restore T-cell immunity encouraged further research.

We further studied the effect of PMZH knockdown SLC43A2 on methionine content in vivo and anti-tumor immunity. We first evaluated the effects of PMZH on SLC43A2 expression in 4T1 tumor-bearing mice. Sanger sequencing of tumor tissue verified that PMZH treatment cause significant mutations in target genes (Fig. 4f). From the immunoblotting experiment, it can be seen that the *SLC43A2* gene was effectively knocked down in the tumors treated with PMZH (Fig. 4g, h). There was no significant change in the expression of SLC43A2 in the liver and kidney tissues of the PMZH group after treatment (Supplementary Fig. 11a, b). We detected the methionine

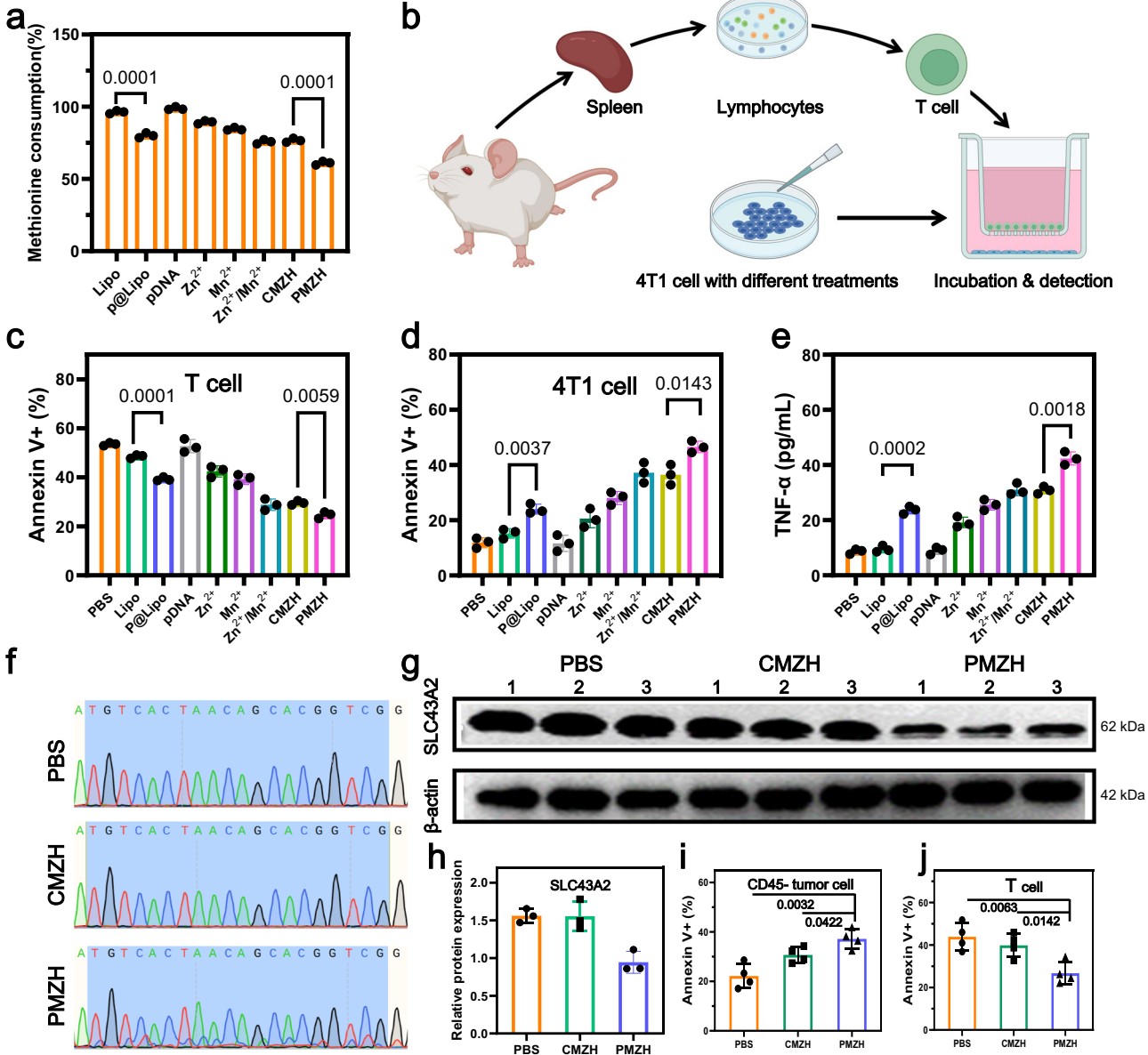

**Fig. 4 | Effects of methionine metabolism regulation on T cell immunity.**
**a** Percentage of methionine consumption in cell culture medium after different treatments. **b** Schematic representation of T cell co-culture with 4T1 cells in the Transwell system. **c** Flow cytometry to detect the changes of T cell viability in different treatment groups. **d** Flow cytometry quantitatively assesses the changes in the viability of 4T1 cells in different treatment groups. **e** Quantitative analysis of cytokines in the culture medium of different treatment groups. **f** Sanger sequencing of PCR amplicons of target loci after different treatment in tumor. **g** Western

blot analysis of SLC43A2 protein expression in tumors after indicated treatments. **h** Semi-quantitative analysis of SLC43A2 expression in tumors. Effect of different treatments on apoptosis of **i** tumor cells and **j** tumor-infiltrating CD8$^+$ T cells in vivo. $n = 3$ independent samples in **a**, **c**, **d**, **e**, **g**, **h**; $n = 4$ independent samples in **i**, **j**. Representative results showed in **f** were obtained from three independent samples. Data were presented as mean ± SD. *P* values were assessed using Student's *t* test (two-tailed). Source data are provided as a Source Data file.

content in serum, liver, kidney, and tumor of different groups of mice (Supplementary Fig. 11c), and further evaluated the apoptosis of tumor cells (CD45⁻) and infiltrating CD8$^+$ T cells in different groups of tumors (Fig. 4i, j and Supplementary Fig. 12). PMZH significantly reduced the apoptosis rate of cytotoxic T cells compared with CMZH, indicating that the knockdown of SLC43A2 was helpful for the survival of T cells. All these results illustrated that selective targeting tumor methionine metabolism can efficiently restore T cell immunity and improve cancer immunotherapy.

**PMZH activated the STING signaling pathway**
In addition to the regulation of methionine metabolism initiated by the CRISPR/CAS9 system, the released large amounts of manganese and

zinc ions also played a crucial role in the antitumor immunity of PMZH. Large amounts of Mn$^{2+}$ and Zn$^{2+}$ would help to enhance cGAS/STING signal transduction and generate a large amount of ROS to damage mitochondria and induce tumor cell death. We assessed intracellular ROS production after different treatments by 2′, 7′-dichlorodihydro-fluorescein diacetate (DCFH-DA) indicator. Flow cytometry (Fig. 5a) results showed that a large amount of ROS was detected in PMZH-treated 4T1 cells, which was superior to ZIF-8@HA, MZH, and CMZH-treated groups. This suggested that bimetallic ions induced ROS production more efficiently, and methionine metabolic regulation also increased intracellular ROS production, possibly due to the increased sensitivity of cells to oxidative stress by methionine restriction. A large amount of ROS generated in cells damaged mitochondria, leading to

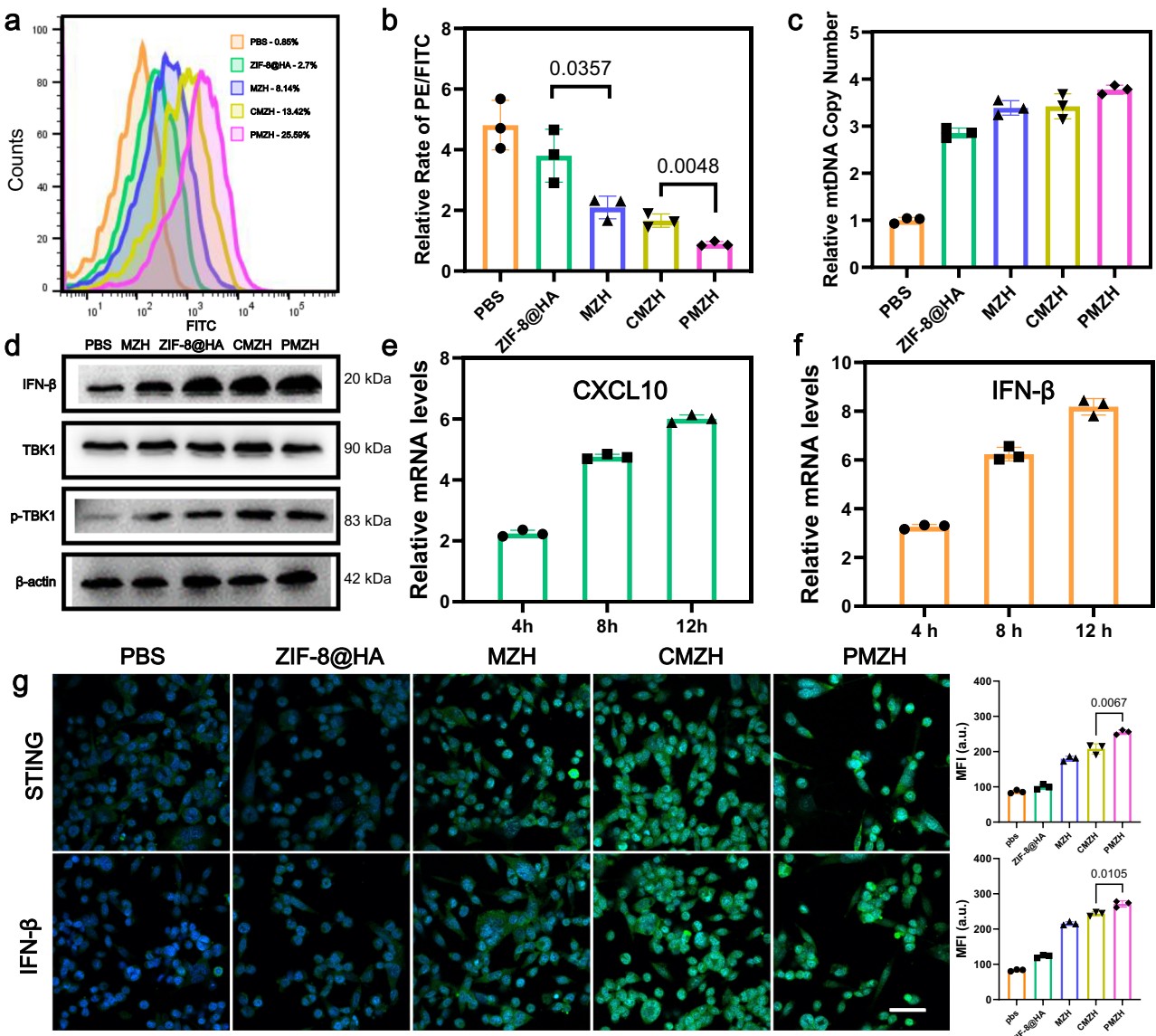

**Fig. 5 | PMZH nanoplateform promotes STING signaling activation. a** ROS generation in 4T1 cells assessed by flow cytometry after different treatments. **b** Mitochondrial membrane potential's changes in 4T1 cells after indicated treatments were evaluated by flow cytometry. **c** MtDNA copy numbers in 4T1 cells after different treatments were detected by real-time PCR. MtDNAs were normalized to β-actin mRNA encoded by the nuclear gene. **d** Immunoblot analysis of important proteins in the STING pathway in 4T1 cells after corresponding treatments. The expression of cGAS target genes **e** CXCL10 and **f** IFN-β in 4T1 cells was assessed by real-time PCR after different time periods of PMZH treatment. **g** Immunofluorescence staining of 4T1 cells with the indicated treatments. (Scale bar: 50 μm). *n* = 3 independent samples in **b**, **c**, **e**, **f**, **g**. Representative results showed in **a**, **d** were obtained from three independent samples. Data were presented as mean ± SD. *P* values were assessed using Student's *t* test (two-tailed). Source data are provided as a Source Data file.

the release of mtDNA, which enhanced cGAS/STING signaling activation. We assessed mitochondrial transmembrane potential (ΔΨm) and the amount of mtDNA. The JC-1 probe was used to detect ΔΨm. JC-1 are red fluorescent J-aggregates when Ψm is high while JC-1 are green fluorescent monomers when Ψm is low. We assessed the extent of mitochondrial depolarization by the ratio of red/green fluorescence intensity. 4T1 cells treated with ZIF-8@HA, MZH, CMZH, and PMZH presented decreased ΔΨm (Fig. 5b). The ΔΨm decreased the most when treated with the incubation of PMZH. We further examined the amount of mtDNA. Different treatment groups all increased the release of mtDNA, and the PMZH group released the most (Fig. 5c). We evaluated the cell viability of different treatment groups through MTT experiments (Supplementary Fig. 13a, b), and the results showed that PMZH had a good antitumor effect. We further assessed the ability of different treatments to activate the STING signaling pathway. The

enhanced activation of the STING signal stimulates the phosphorylation and translocation of TANK-binding kinase 1 (TBK1) into the nucleus and produces type I IFN to enhance antitumor effects [69,70]. We first performed immunoblot analysis to confirm the expression of p-TBK1 and the important cytokine IFN-β in the cGAS/STING pathway (Fig. 5d). IFN-β is a hallmark cytokine of STING pathway activation, and it plays a crucial role in the activation of subsequent innate immunity and the connection of adaptive immunity. As shown in Fig. 5d, the PMZH treatment group powerfully upregulated the p-TBK1 and IFN-β expression. The expression of CXCL10 (an important pro-inflammatory chemokine in the STING signaling pathway and T cell infiltration) and IFN-β in the culture medium of 4T1 cells was then assessed using qPCR after PMZH treatment various times (Fig. 5e, f). The results displayed that PMZH significantly upregulated the mRNA levels of IFN-β and CXCL10. These indicators have been widely used to

evaluate the activation of STING pathway. The results of cellular immunofluorescence experiments (Fig. 5g) further proved the promotion of STING pathway activation by PMZH. Therefore, we believed that PMZH could effectively induce ROS and trigger the mitochondrial apoptosis pathway to kill tumor cells. During this process, a large amount of mtDNA was released into the cytoplasm, and synergistically enhanced STING signaling with nutrient metal ions, triggering powerful antitumor immunity.

Activation of the STING signaling pathway in antigen-presenting cells is also crucial for antitumor immunity. Hence, we evaluated the STING pathway activation in bone marrow-derived dendritic cells (BMDCs) in vitro after treatment by different groups (Supplementary Fig. 14a, b). It can be observed that more cytokine IFN-β and CXCL10 production was found in the supernatant of the BMDC medium after incubation with PMZH, implying potent enhanced the STING signaling in BMDCs. Due to the enhancement of the STING signaling in BMDCs and subsequent IFN-β production contributing to BMDCs maturation, we performed a further evaluation of PMZH-treated BMDC cells at different concentrations. Flow cytometry results showed that PMZH promoted the maturation of BMDCs in a dose-dependent manner (Supplementary Fig. 14c, d). Treatment of PMZH would help the maturation of BMDCs and secretion of IFN-β, thereby enhancing systemic antitumor immunity.

## In vivo antitumor effect of PMZH

Next, we further evaluated the therapeutic effect of PMZH in vivo. Before that, we first evaluated its security. The low hemolysis rate of PMZH indicated its good blood compatibility and could be safely used for injection (Supplementary Fig. 15). Then, we assessed the short- and long-term biocompatibility of PMZH in vivo. Compared with PBS group, there was no obvious change in the mice body weight of the PMZH group (Supplementary Fig. 16b). All indicators of blood biochemistry, hematology, liver status, and kidney function did not change obviously, illustrating that there was no obvious infection and inflammation in PMZH treatment (Supplementary Fig. 16a). In addition, morphological differences and damage in major organs were not observed by hematoxylin and eosin (H&E) staining assays (Supplementary Fig. 16c). Effective accumulation of PMZH in tumors is a prerequisite for achieving desirable antitumor efficacy. We assessed the biodistribution of PMZH in vivo using a near-infrared fluorescence imaging system and ICP analysis. We collected mice tumors and major organs at various time after intravenous injection of PMZH for assessment. ICP analysis showed that PMZH could be efficiently enriched at tumor sites and metabolized over time (Supplementary Fig. 17a). In near-infrared fluorescence imaging studies, after intravenous injection of DiD-labeled PMZH and Cy5-labeled PMZH, a strong fluorescence signal was observed in the tumor, which also verified the efficient tumor accumulation of PMZH (Supplementary Fig. 17b, c). The tumor targeting effect of HA enhanced the enrichment at the tumor site (Supplementary Fig. 17d). In vivo MRI also indicated the accumulation of PMZH in the tumor (Fig. 6h).

Next, we evaluated the antitumor effects of PMZH. We randomly divided 4T1 tumor-bearing mice into the PBS group, ZIF-8@HA group, MZH group, and PMZH group. We started different groups of treatments when the tumor volume was ~60–70 mm$^3$ and recorded changes in the tumor volume and mice's body weight during the treatment period (Fig. 6a). No significant body weight changes were observed in the PMZH group throughout the treatment period, suggesting negligible systemic toxicity of PMZH (Fig. 6c). Compared with the PBS group, the PMZH group exhibited an highly inhibitory effect on tumor growth (Fig. 6d–g and Supplementary Fig. 18). The tumor growth inhibition rate (TGI) of the PMZH group reached 85.47%, which was much higher than that of the other groups (Fig. 6e). Furthermore, mice exhibited considerable survival after PMZH treatment (Fig. 6b). The tumor H&E staining after different treatments also demonstrated the

good tumor-killing effect of PMZH (Fig. 6i). These all demonstrated that CRISPR plasmid-encapsulated bimetallic MOF has a strong antitumor effect, which may be attributed to the synergistic effect of oxidative stress and methionine metabolism regulation. This was also demonstrated by ROS evaluation of tumor sections from different groups (Fig. 6j) and methionine metabolism assessment experiments described earlier.

## In vivo antitumor immunity effect of PMZH

After different treatments, we took the tumors and lymph nodes of mice and further assessed the infiltration level of antitumor immune cells by flow cytometry. Free Mn$^{2+}$/Zn$^{2+}$ and mtDNA would promote STING pathway activation and dendritic cells (DC) maturation. DC maturation is critical for T cell activation. Flow cytometry analysis of lymph nodes showed that ZIF-8@HA group just moderately promoted DC maturation, while the percentage of activated DCs was significantly increased in the MZH and PMZH groups, and PMZH group showed the highest (Fig. 7a). This suggested that the involvement of multiple nutrient metal ions and enhanced mtDNA release were more favorable for STING pathway activation and DC maturation. STING-mediated DC maturation further promoted the initiation and activation of T cells. The two main types of effector T cells are CD8$^+$ T cells and helper T lymphocytes (CD4$^+$ T cells). CD8$^+$ T cells can directly kill cancer cells, while CD4$^+$ T cells function to regulate other immune cells. The percentage of CD8$^+$ T cells was increased in different treatment groups, and the PMZH group reached the highest (Fig. 7b), indicating that PMZH effectively increased CD8$^+$ T cell infiltration. The percentage growth of CD4$^+$ T cells was similar to that of CD8$^+$ T cells (Fig. 7c). We collected serum samples from different treatment groups and assessed their cytokine (IL-6 and TNF-α) content by ELISA (Fig. 7d-e). The highest levels of cytokines were observed in the PMZH group, further suggesting that PMZH effectively enhanced T cell infiltration and function, activating a robust immune response. Not only in the 4T1 tumor model, but also the CT26 tumor model, PMZH has the ability to enhance anti-tumor immunotherapy (Supplementary Fig. 19). The expression of IFN-β detected by real-time quantitative PCR was similar to the above results, and PMZH effectively increased the expression of IFN-β significantly (Fig. 7f). We further evaluated the effects of different treatments on the cGAS/STING pathway in 4T1 tumor-bearing mice. In each group, three tumor samples were randomly selected for protein extraction, and immunoblotting experiments were performed. It can be seen, that the PMZH-treated group effectively activated cGAS/STING signaling (Fig. 7g, h). At the same time, immunoblotting experiment also confirmed that methionine restriction caused by PMZH contributed to the activation of STING pathway (Supplementary Fig. 20).

Then, we constructed distant tumor models (Fig. 8a) and lung metastasis models (Supplementary Fig. 21a) to simulate tumor relapse and metastases. We inoculated the distant 4T1 tumors on the other side after treatment firstly. PMZH could provide long-term protection against tumor recurrence due to the improved immune response after PMZH treatment. Compared to other treatment groups, the distant tumor growth was significantly inhibited in mice whose primary tumors were treated with PMZH (Fig. 8b–d and Supplementary Fig. 22), and the TGI was 87.78% (Fig. 8e). We detected CD8$^+$ T cells infiltration in the distant tumors by immunofluorescence staining. Immunofluorescence images showed that the distant tumor areas in the PMZH-treated group had a large number of CD8$^+$ T cells infiltration, meaning that tumor-specific immunity was activated (Fig. 8g). This was also confirmed by the presence of large areas of apoptosis and necrosis in H&E staining of distant tumors treated by PMZH (Fig. 8f). The 4T1 breast tumor is highly aggressive for lung metastasis, so we further evaluated the efficacy of PMZH by constructing lung metastasis models. Compared with the PBS treatment group, lung tissue photos of PMZH group mice did not show obviously metastatic

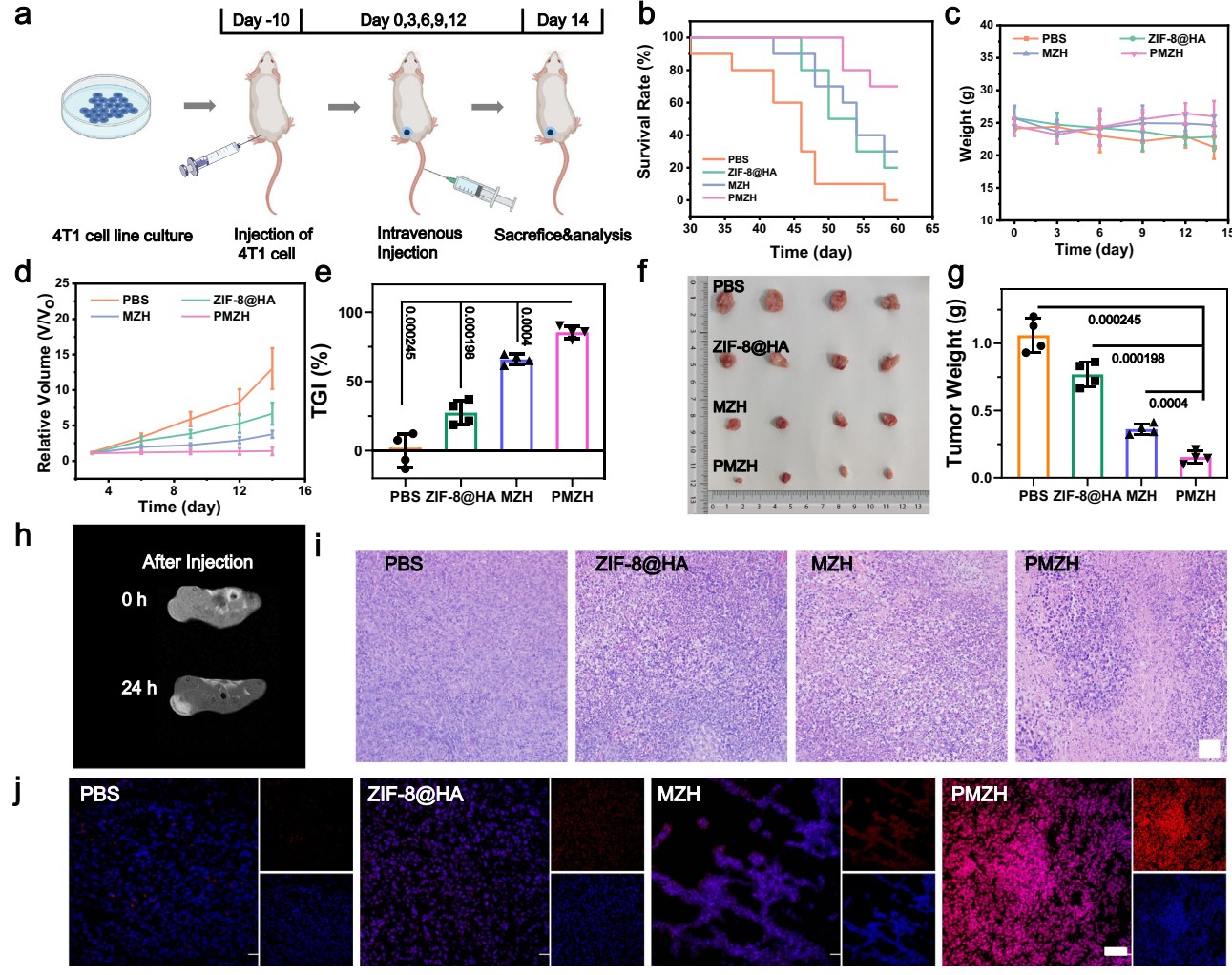

**Fig. 6 | Antitumor therapeutic effect of PMZH in vivo. a** Schematic representation of vaccination and treatment. **b** Survival rates (*n* = 10 mice per group), **c** Body weights, **d** tumor volume changes, **e** TGI rates, **f** photographs of dissected tumors, **g** tumor weights of mice after different treatments. **h** MRI images after PMZH injection in 24 h. Data were presented as mean ± SD (*n* = 4 mice) in **c, d, e, g**. *P* values were assessed using Student's *t* test (two-tailed). **i** H&E staining pictures of tumors after corresponding treatments. **j** ROS evaluation in tumors after different treatments. *n* = 3 mice per group in **h**–**j**. Source data are provided as a Source Data file. (Scale bar = 50 μm in **i** and 100 μm in **j**.).

nodules (Supplementary Fig. 21b). H&E staining exhibited that compared with the PMZH group, the lung sections of PBS treatment groups showed higher degree of canceration (Supplementary Fig. 21c). These experimental results demonstrated that PMZH promoted systemic immune responses that effectively inhibited tumor relapse and metastasis.

We performed further evaluations of the mechanism of PMZH affecting T cells. We assessed T cell viability and function after different treatments in wild-type 4T1 tumor-bearing mice and in STING-KO 4T1 tumor-bearing mice (Supplementary Fig. 23). The results demonstrated that the enhancement of T cell function by PMZH was more pronounced in wild-type 4T1 tumor-bearing mice. All the results indicated that the activation of the STING pathway increased T cell infiltration, while the regulation of methionine metabolism restored T cell function. The synergistic effect of both effectively enhanced T-cell immunity.

## Discussion

In summary, we construct a nanoplatform that can increase CD8+ T cell infiltration and enhance CD8+ T cell function. The CRISPR plasmid targeting *SLC43A2* restricted methionine metabolism in tumor cells and enhanced T cell immunity by relieving the methionine

competition pressure of CD8+ T cells. Nutrient metal ions induced ROS production synergized with methionine restriction, thereby stimulating cGAS/STING pathway activation and increasing T cell infiltration. Therefore, our work not only effectively activated the cGAS/STING pathway but also enhanced T cell immunity by regulating metabolism, thereby activating the robust antitumor immune response to inhibit tumor recurrence and metastasis, providing clinical potential for promoting antitumor immunotherapy.

## Methods

### Cell lines, mice and ethical statement

The cell lines (4T1 cell line, Catalog no. CRL-2539; CT26 cell line, Catalog no. CRL-2638) were provided from ATCC and all the cell lines authenticated by ATCC before provided. The supplier stated that the test results for Mycoplasma contamination in these cell lines were negative. All cell lines used in this work are not listed as cross-contaminated or misidentified cell lines (v12, 2023) by International Cell Line Authentication Committee.

In all, 200 female BALB/c mice (22–27 g, 6–8-week-old) needed for the experiment were purchased from the Medical Experimental Animal Center of Jilin University (Changchun, China). The handling procedures were in compliance with the guidelines of Jilin University Animal

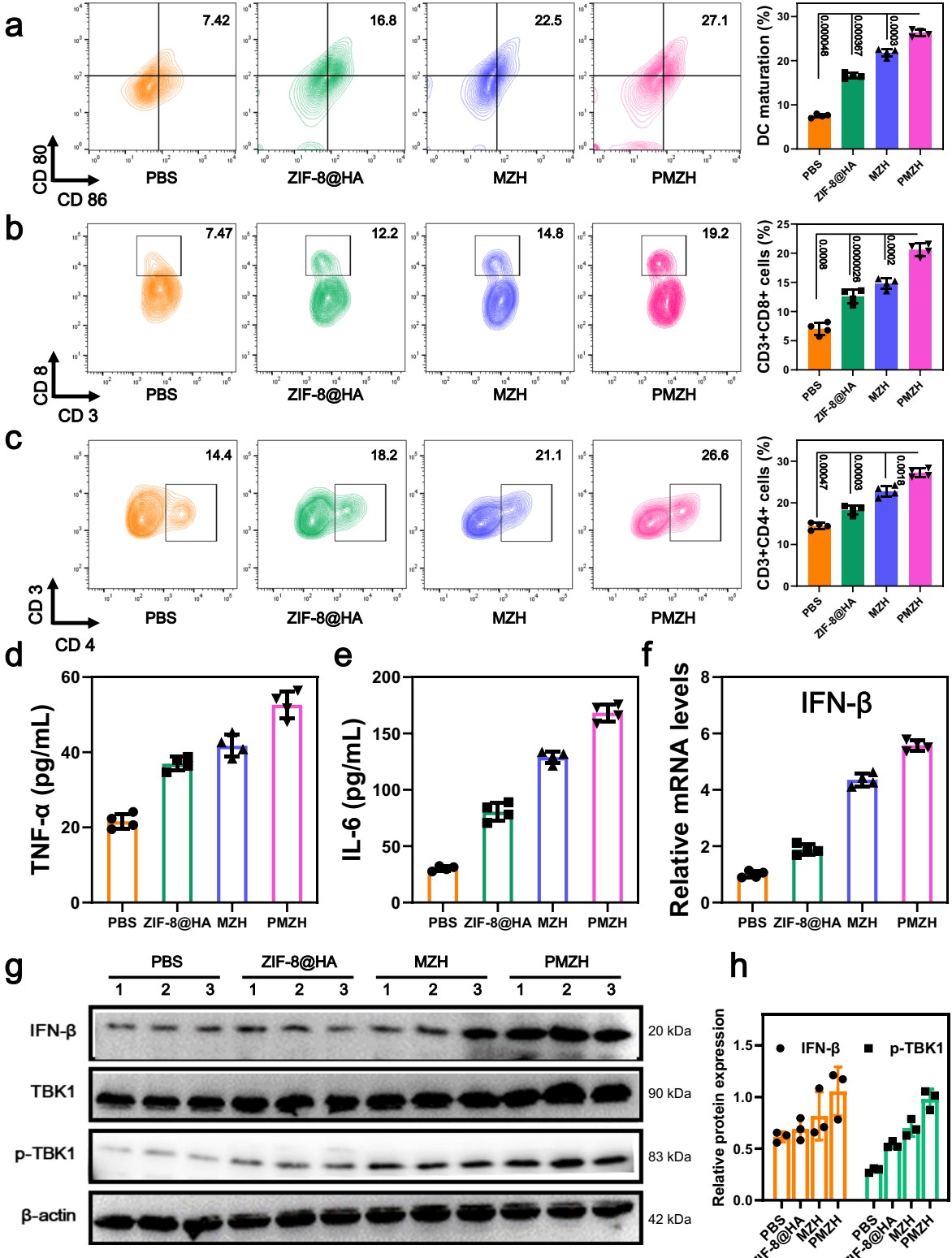

**Fig. 7 | Immunity therapy of PMZH. a** Flow cytometry analysis of DCs (gating on CD11c+) infiltrated after treatment with different formulations. Populations of **b** CD8+ and **c** CD4+T cells in 4T1 tumor tissues after different treatment. Cytokine expression levels of **d** TNF-α and **e** IL-6 in serum were analyzed by Elisa kit after different treatments. **f** The IFN-β expression in tumors treated with different preparations was detected by qPCR. Data were presented as mean ± SD ($n = 4$ independent samples) in **a**–**f**. $P$ values were assessed using Student's $t$ test (two-tailed). **g** Immunoblotting analysis of STING pathway-related protein levels in tumor samples from various treatment groups. **h** Semi-quantitative analysis of p-TBK1 and IFN-β expression. $n = 3$ independent samples in **h**. Data were presented as mean ± SD. Source data are provided as a Source Data file.

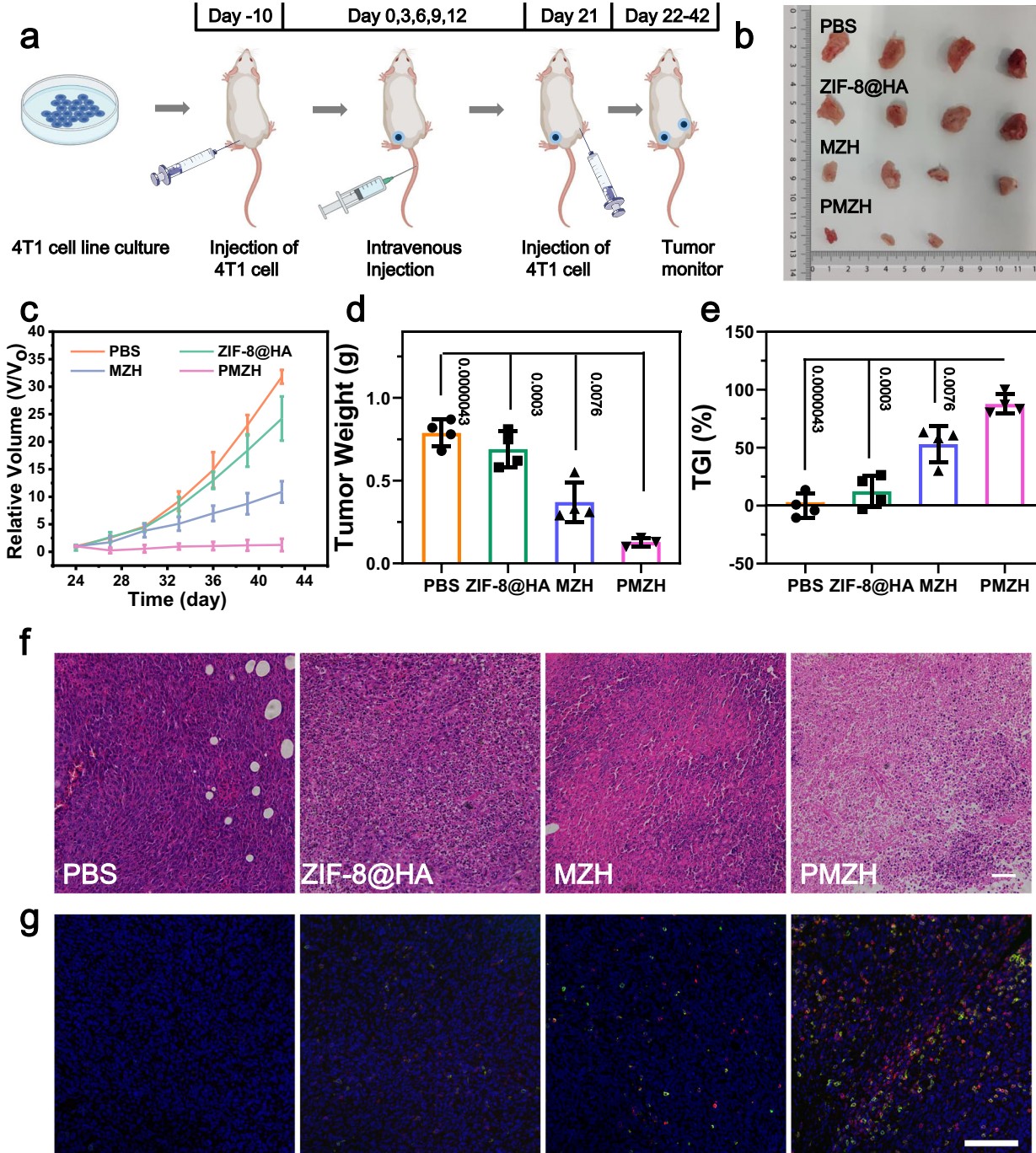

**Fig. 8 | Evaluation of antitumor immune effects of PMZH on distant tumors.**
**a** Schematic diagram of inoculation time and treatment time. **b** Photograph of the distant tumors in different treatment groups. **c** Relative volume change of distal tumor, **d** distal tumor weight, and **e** TGI rate of the distant tumors. Data were presented as mean ± SD ($n = 4$ mice). $P$ values were assessed using Student's $t$ test (two-tailed). **f** H&E staining images of distal tumor. (Scale bar: 50 μm). **g** Immunofluorescence staining of CD8+ T cells in distal tumor. CD3+ is stained in red fluorescence while CD8+ is stained green fluorescence. (Scale bar: 100 μm). $n = 3$ mice per group in **f**, **g**. Source data are provided as a Source Data file.

Care and Use Committee. All mice were housed at 12-h light-dark cycle within 25–27 °C and 45–55% humidity. In our work, no tumor burden exceeds 2000 mm³, which meets the maximum tumor burden permitted by the protocol of the Animal Care and Use Committee of Jilin University. Tumor volume was calculated as: width²×length/2.

### Engineering of CRISPR/Cas9 system
To select the optimal sgRNA for efficient genome editing, three pairs of oligonucleotide sequences were designed using the website tool (https://zlab.bio/guide-design-resources) based on the analysis of

genomic sequence of *SLC43A2*. The sequences were ligated into the BbsI restriction site of pX330 (Addgene) for generating the sgRNA expressing vector (CRISPR/Cas-S). The sgRNA (5′-CACCGGTGTCAC-TAACAGCACGGTCG-3′) targeting *SLC43A2* with satisfactory efficiency were selected as the optimal candidates for constructing the CRISPR/Cas9 system. Direct sequencing was conducted after acquiring the amplified PCR products. A large number of target plasmids were then obtained by amplification in competent cells. SPARKeasy High Purity Plasmid Small Quantity Rapid Extraction Kit was purchased from Shandong Sparkjade Biotechnology Co., Ltd.

## Synthesis of pDNA@Mn/Zn − ZIF (PMZ)

First, we mixed 30 μL of aqueous solution (DNase free) containing pDNA 30 μg and 125 μL of aqueous solution containing dimethylimidazole 23.75 mg well at room temperature condition and stirred for 5 min. Next, 125 μL aqueous solution containing $Zn(NO_3)_2·6H_2O$ 2.4 mg was added and stirred for 10 min. After stirring, the product (pDNA@ZIF-8, PZ) was collected by centrifugation. After washing twice, freeze drying it for further use. Then, we synthesized pDNA@Mn/Zn-ZIF: First, we fully dissolved 3 mg of manganese acetyllacetonate in 100 μL methanol (MeOH), then we added 1 mg pDNA@ZIF-8 to it. The solution was incubated in an oven at 55 °C for 24 h. The product (denoted as pDNA@Mn/Zn−ZIF, PMZ) was collected and washed with methanol by centrifugation at $10,000 \times g$ several times until the supernatant turned colorless. Then, the nanoparticles obtained from the previous step were soaked in methanol for 3 days, and the original solution was replaced daily with a new methanol solution (100 μL). It was collected by centrifugation after 3 days and dried for further use. Mn/Zn-ZIF (MZ) was synthesized in a similar manner.

## Synthesis of PMZH

In total, 15 mg PMZ were first dispersed in 10 mL of ultrapure water, then 10 mL of PAH (10 mg) solution was added. After stirring for 2 h at 4 °C, we centrifuged and washed to obtain the nanoparticles. 2 mL HA solution (12 mg) was added and stirred overnight at 4 °C. Mn/Zn-ZIF@HA (MZH) was synthesized in a similar manner.

## In vitro degradation behaviors of PMZH

PMZH was added to PBS buffer (pH = 7.4 and 6.5) respectively. Then incubated the test solutions in a 37 °C water bath with stirring. Calculated the ion release rate at different times by ICP-MS.Acid-responsive release of pDNA was assessed by agarose gel electrophoresis.

## Agarose gel retardation assay

In all, 8 μL nanomaterials aqueous solution at different binding ratios were mixed with 2 μL 5× DNA loading buffer to obtain different loading solutions. Then, added the above loading solution, the DNA leader, and free pDNA loading solution to the loading wells of a 1% agarose gel containing Serred and imaged after electrophoresis under 80 V conditions 0.5 h.

## Analyses of cellular uptake and endosomal escape

4T1 cells were seeded in plates, and after incubation for overnight, FITC-modified PMZH was added to incubate for additional hours. After that, cells were collected by digestion and centrifugation, the uptake of nanoparticles by cells was evaluated by flow cytometry.

For endosome escape capability analysis, 4T1 cells were seed into plates and cultured. Then incubated with the YOYO-1-labeled nanoparticles for another 0.5,1, 2, and 4 h. Thereafter, LysoTracker Red and DAPI were used for staining the lyso/endosomes and nuclei at designed time points, respectively. The cells were rinsed and fixed, and then observed under CLSM.

## Gene transfection

To investigate the transfection efficiency of the prepared nanoparticles, 4T1 cells were seeded into plates prior to transfection. The PMZH and lipo-6000 containing same amount of pEGFP were then added into cells, and then cultured in FBS-free medium for 6 h. Hereafter, replaced with fresh DMEM medium and cultured for another day. Lastly, the collected cells for transfection efficacy were evaluated by the fluorescence microscope and flow cytometry.

## Western blot assay

After different formulations treated, cells were collected and harvested using RIPA Lysis Buffer to obtained proteins. Use the BCA protein detection kit (Sangon Biotechnology Co., Ltd.) for protein quantification and dilute it to the same concentration. These proteins were then separated by SDS-PAGE gradient gel and transferred to PVDF membrane. Then, the membranes were incubated with antibodies against TBK1 Rabbit Polyclonal Antibody, Phospho-TBK1/NAK (Ser172) Rabbit Polyclonal Antibody, IFN-Beta Antibody, SLC43A2 Antiboday, and β-actin at 4 °C overnight, followed by an HRP-conjugated goat anti-mouse immunoglobulin G (IgG). The protein bands were performed using the enhanced chemiluminescence solution reaction. The detailed antibody information is listed in Supplementary Table 1. We have provided uncropped and unprocessed scans of the most important blots in the source data file.

## RNA extraction and RT-qPCR

RNAs were isolated according to the protocol of TRzol reagent (Beyotime) with minor modification. cDNA was synthesized using RT reagent kit with gDNA Eraser (Beyotime). SYBR Green Abstart PCR Mix (Sangon Biotechnology Inc.) was used to perform qPCR on Toptical 96 Real-Time PCR System. β-actin were used as endogenous controls. The detailed primer information is listed in Supplementary Table 2.

## In vivo biocompatibility

BALB/c mice were randomly divided into three groups. After 0, 7, and 28 days of intravenous injection of PMZH, their blood was collected for blood biochemical analysis, and their main organs were extracted for H&E staining analysis.

## In vivo anti-tumor therapy/in vivo anti-tumor immunotherapy

Subcutaneously injected 4T1 cells into mice and waited for tumor growth. When the tumor volume of the mice increased to 75-100 $mm^3$, they were divided into four groups randomly as follows: (1) control injected with saline, (2) ZIF-8@HA, (3) MZH (4) PMZH. Injections (0.625 mg/mL,100 μL) were performed once every three days. The tumor volumes and body weights of mice were measured every 3 days.

## Evaluate the T cell viability in vitro

Mouse lymphocytes were isolated from spleen by Mouse Lymphocyte Separation Medium.

Re- suspended CD8 + T cells ($10^6$ cells/ml) and activated with anti-CD28 and anti-CD3 mAbs for 48 h. The activated CD8 + T cells were maintained with fresh complete medium contained 2-mercaptoethanol and IL-2. 4T1 cells with different treatment and CD8 + T cells were co-cultured in a Transwell system for 72 h, and then evaluate the viability of CD8 + T cells by flow cytometer.

## In vivo flow cytometry assay in tumor issue

After therapy, the mice were euthanized to obtain tumor tissue. Obtain a single-cell suspension by grinding the tumor and filtering. After being treated with Red Blood Cell Lysis Buffer (Solarbio), the single cells were then collected and washed with PBS and stained with corresponding antibodies following the standard protocol. Following staining, cells were washed, fixed, and then analyzed by flow cytometer.

## Lung metastasis model

The untreated mice and PMZH cured mice were intravenously injected 4T1 cells. After 20d, lung sections were harvested from each group and further studied by H&E staining assay.

## Statistical analysis

Data were presented as mean ± SD ($n$ = 3 unless specified.).Statistical analysis was performed using Microsoft Excel, Origin 2020, and GraphPad Prism software using Student's $t$ test to compare statistical significance between two groups. We calculated the p-value through the student's t-test and labeled it in the graph. Representative results showed in Figs. 2b−e; 3a, c, d, f−h; 4f, g; and 5a, d and Supplementary

Figs. 1; 3a, b; 5; 6; and 21c were generally obtained from three independent samples unless specified.

## Reporting summary

Further information on research design is available in the Nature Portfolio Reporting Summary linked to this article.

## Data availability

The source data for Figs. 2–8 and Supplementary Figs. 2–5, 7–11, 13–20, 22, and 23 are provided in this paper. The remaining data are available within the Article, Supplementary Information, and Source Data file. Source data are provided with this paper.

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

## Acknowledgements

This work was supported by the National Key R&D Program of China (2019YFA0709202, X.Q. and 2021YFF1200700, J.R.), National Natural Science Foundation of China (21820102009, X.Q., 91856205, X.Q., 22237006, J.R., 22107098, G.Q., and 22122704, C.Z.), and Jilin Innova-tion Project (2023DJ02, X.Q.).

## Author contributions

J.R. and X.Q. conceived and designed the experiments, supervised the study and revised the manuscript. Y.H. performed the experiments, analyzed the data, and wrote the original manuscript; G.Q. and C.Z. performed the experiments, analyzed the data, and discussion. T.C. performed the experiments.

## Competing interests

The authors declare no competing interests.
