## [Peer Review File · Nature Communications]

A bimetallic nanoplatform for STING activation and CRISPR/Cas mediated depletion of the methionine transporter in cancer cells restores anti-tumor immune responsesREVIEWER COMMENTS

Reviewer #1 (Remarks to the Author): with expertise in cancer nano-therapy

In this manuscript, Huang et al. constructed a synergistic nanoplatfrom to regulate methionine metabolism and activate the STING innate immune pathway. The author combined methionine metabolism, CRISPR system and nutrient metal ions to alter tumor microenvironment and enhance immunity. This system was tested in vitro under various conditions, and was further demonstrated to have significant efficacy in mouse breast cancer and mouse colorectal cancer animal models. Their results are interesting and show promising for biomedical applications. So, I would recommend this manuscript for publication after addressing the following minor concerns :

1. There are mainly two mechanisms of the nanoplatfrom constructed by the article. On one hand, modulation of methionine metabolism by CRISPR plasmids restores T cell function. On the other hand, methionine metabolism cooperates with nutrient metal ions to activate the sting immune pathway. However, the current description of the effects of methionine metabolism on the sting pathway is confusion. Please adjust the sequence of picture combinations to a more systematic description of the activation of the sting pathway. Moreover, it is not sufficient to introduce a CMZH control group in vivo only. Complementing relevant experiments at the cellular level is necessary.
2. The authors did not adequately discuss the superiority for using a gene editing approach to inhibit SLC43A2; siRNA for example may work better considering the reduced delivery barriers.
3. Because PMZH is pH sensitive, was the dye released from PMZH and escaped from endo/lysosomes? The author needs a further explanation for this.
4. The numerical changes in ROS detection in figure4a are not given, please supplement.
5. The authors claimed that modification with HA could enhance targeting effect, however, relevant data was not represented in the manuscript.

Reviewer #2 (Remarks to the Author): with expertise in cancer immunology, metabolism

Tumor immunotherapy based on nanoparticles is a new therapy developed rapidly in recent years. Many nanoplatfoms, such as Iron-based nanoscale frameworks encapsulated with CRISPR/Cas9 have been reported (J. Am. Chem. Soc. 2018, 140). In this manuscript, the authors developed a nanoplatfom containing hyaluronic acid, CRISPR/Cas9 plasmid and metal ions, and demonstrated its antitumor activity. The hyaluronic acid modification was used to enhance tumor recognition and accumulation. CRISPR/Cas9 system down-regulated the methionine transporter of tumor cells to restrict the methionine uptake of tumor cells, which relieves the methionine competition pressure of CD8+ T cells and restores the function of tumor-infiltrating CD8+ T cells. Lastly, the nutrition metal ions (Mn^{2+} and Zn^{2+}) could not only induce ROS storm to kill tumors but also stimulated the activation of the STING pathway. The design is interesting and each component plays a unique role, which could work in a cooperative manner. Overall, the findings are interesting, but extensive experiments are required to frame the results.

Major concerns:

- (1) In this study, the authors claimed that the nanoparticles could release ions in acid lysosome conditions and catalyze intracellular H_2O_2 to ROS to kill cancer cells. However, the tumor microenvironment (TME) is also acidic (pH 6.5-6.8) with a high concentration of H_2O_2 . Are the nanoparticles stable in TME? Will they also catalyze TME H_2O_2 to ROS? Why does they affect T cell functions?
- (3) Why the PMZH effectively knocked down SLC43A2 in tumors, but not in liver and kidney? The biodistribution of this nanosystem in organs should be performed.
- (4) The ROS increase upon nanoparticles accumulation in the tumor should be confirmed in vivo.
- (5) It has been known that Mn^{2+} dramatically promotes pro-inflammation response and anti-tumor immunity dependent on macrophages. Would the nanoparticles affect other immune cells (such as macrophages) and remodel the tumor immune microenvironment? How to demonstrate the nanoparticles specifically promotes CD8+ T cells antitumor function rather than the changes of the immune microenvironment?
- (6) The authors claimed that PMZH remodels methionine metabolism in T cells. However,

the authors don't provide any data to support that intervention of methionine metabolism in tumor cells could affect T cell immunity in this nanosystem.

(7) lipo has cytotoxicity in cells. However, the authors showed that the lipo treatment in Figure 3C decreased the apoptosis of T cells. Please explain.

Minor concerns:

(1) The full name of ZIF should be spelled out in the manuscript.

(2) The quantitative colocalization analysis of plasmid with endo/lysosomes in 4T1 cells should be conducted. The color scatter plots and corresponding Pearson's correlation coefficient (PCC) values between the red and green fluorescence signals in the images should be provided.

(3) The title of the y-axis in Figure 4A should be labeled.

(4) The quality of WB results should be improved (such as fig2f, fig6g).

Reviewer #3 (Remarks to the Author): with expertise in cancer nano-therapy

In this paper, Huang et.al constructed a synergistic nanoplatform by encapsulating CRISPR plasmids into Mn/Zn bimetallic MOF nanoparticles. The nanoplatform promoted CD8 + T cell infiltration in tumors while restoring T cell function by modulating methionine metabolism and activating the sting innate immune pathway. This study is based on previous work of Weiping Zou (nature volume 585, pages277 – 282 (2020)) to exploit differences in methionine metabolism to design protocols to enhance T cell infiltration and restore T cell function . It is a topic of interest to researchers in cancer treatment, biomaterials, and other related fields. Some minor revisions should be considered before publication in Nature Communications.

1. The authors introduced a CMZH control group in their study in vivo to exclude the effect of a nonfunctional plasmid on methionine metabolism. However, they did not provide the effect of the CMZH control group on SLC43A2 gene expression. Please supplement. In addition, compared with CMZH, PMZH effectively reduced the level of apoptosis of T cells, but the effect of PMZH and CMZH on the secretion of functional cytokines was not evaluated. This needs further research and discussion.

2. Cytosolic DNA can be recognized by CGAs and other DNA sensors to produce type I interferons and antitumor innate immunity. Although the authors demonstrated the effects of methionine transport on the sting pathway in vivo, experiments at the cellular level are lacking. The CMZH group needs to be introduced in cytological experiments as well.
3. Please supplement the fluorescence intensity scale bar of biodistribution (supplement, Figure10b).
4. Flow gate strategies for DC cells, Figure7f-g should be provided in SI.
5. The targeting effect of HA needs to be characterized.
6. The authors should clarify why they choose plasmid DNA but not mRNA to encode the CRISPR/Cas9. Usually the immunogenicity of mRNA is much lower than plasmid DNA and can achieve higher protein expression in vivo.
7. Please check for formatting issues throughout this manuscript.

Reviewer #4 (Remarks to the Author): with expertise in cancer immunology, metabolism

Here the authors generate nanoparticle formulations that allow for encapsulating a plasmid encoding for sgRNA targeting SLC43A2 and CRISPR/Cas9. Using this platform, the authors show that reduction of SLC43A2 in 4T1 tumors results in increased T cell function in vitro and in vivo (e.g. tumor killing/rejection) and this is due to activation of the STING/cGAS pathway.

My major comment is related to an unspecified mechanism of action of this nanoparticle. On the one hand, there is reduced tumor methionine consumption and higher T cell function. On the other hand, the enhanced T cell function could be due to a general enhancement in STING/cGAS activation or both. Some genetic loss of function experiments are essential to clarify how these nanoparticles are acting.

A second comment is the use of PBS as a control in all the in vitro experiments. At a minimum there should be a non-targeting sgRNA as well as testing some individual components of the nano-particle itself. Ideally, the authors should do this on the tumor cells alone, T cells alone, and then both in a coculture experiment.

Third, most, if not all, of the micrographs need to be carefully quantified and statistical analysis performed.

Forth, while the authors do examine DC markers and frequency of DCs, careful assessment of STING activation in these DC populations needs to be performed. In other words, does the nanoparticle activate STING in DCs or tumors or another immune population (e.g. myeloid). The authors suggest this happens in DCs but do not show these data.

While I am fine with some methods being deposited in the supplemental information, this is not sufficient for the reader to understand how certain experiments were performed. Therefore, the key methods of the main figures need to be placed in the manuscript. In particular, there needs to more details on the replicates and statistical test for each figure panel embedded directly in the captions.

Next is the summarization of our point-by-point response to the comments raised by the editor and reviewers.

Reviewer #1 (Remarks to the Author): with expertise in cancer nano-therapy

1. There are mainly two mechanisms of the nanoplatform constructed by the article. On one hand, modulation of methionine metabolism by CRISPR plasmids restores T cell function. On the other hand, methionine metabolism cooperates with nutrient metal ions to activate the sting immune pathway. However, the current description of the effects of methionine metabolism on the sting pathway is confusion. Please adjust the sequence of picture combinations to a more systematic description of the activation of the sting pathway. Moreover, it is not sufficient to introduce a CMZH control group in vivo only. Complementing relevant experiments at the cellular level is necessary.

Our response: We appreciate very much for your constructive comments and kind recommendations. We introduced the CMZH group at the cellular level and adjusted the sequence of picture combinations (**Figure 5, Supplement Figure 20**) . As shown in Figure 5, compared to CMZH, the PMZH group significantly enhanced the generation of reactive oxygen species and the release of mtDNA, thereby enhancing the expression of STING signals. Methionine restriction can enhance the sensitivity of tumor cells to oxidative stress. This phenomenon indicated that the CRISPR plasmid regulated methionine metabolism, thereby enhancing ROS production and promoting the activation of the STING signaling pathway. This is consistent with the conclusion of the previous in vivo experiment.

At the same time, we further explored the relationship of methionine metabolism and the activation of STING (**Supplement Figure 23**) . In STING-KO 4T1 tumor bearing mice, PMZH could still restore T cell function, but its effect is weaker than in wild-type tumor bearing mice. This further elucidated the mechanism of PMZH: activation of the STING pathway increased T cell infiltration, while regulation of methionine metabolism restored T cell function. The synergistic effect of both effectively enhanced T cell immunity.

Figure 5. PMZH nanoplateform promotes STING signaling activation. **a** ROS generation in 4T1 cells assessed by flow cytometry after different treatments. **b** Mitochondrial membrane potential's changes in 4T1 cells after indicated treatments were evaluated by flow cytometry. **c** MtDNA copy numbers in 4T1 cells after different treatments were detected by real-time PCR. MtDNAs were normalized to β -actin mRNA encoded by the nuclear gene. **d** Immunoblot analysis of important proteins in the STING pathway in 4T1 cells after corresponding treatments. The expression of cGAS target genes **e** CXCL10 and **f** IFN- β in 4T1 cells was assessed by real-time PCR after different time periods of PMZH treatment. **g** Immunofluorescence staining of 4T1 cells with the indicated treatments. (Scale bar: 50 μ m). *P* values were assessed using Student's *t* test. Source data are provided as a Source Data file.

Supplementary Figure 20. Western blot analysis of STING pathway-related protein levels in tumor samples from different treatment groups. Source data are provided as a Source Data file.

Supplementary Figure 23. (a) Effect of different treatments on apoptosis of tumor infiltrating CD8⁺ T cells in vivo. (b) Cytokine expression levels of TNF- α in serum were analyzed by Elisa kit after different treatments. *P* values were assessed using Student's *t* test (two-tailed) (**P*<0.01, ***P*<0.005, ****P*<0.001). Source data are provided as a Source Data file.

2. The authors did not adequately discuss the superiority for using a gene editing approach to inhibit SLC43A2; siRNA for example may work better considering the reduced delivery barriers.

Our response: We sincerely appreciate the insightful comments of the reviewer. We have added a discussion on this issue in the manuscript. CRISPR/Cas9 has the advantages of

permanently modifying target genes, being exceedingly selective, and having low off-target likelihood. It is one of the most efficient, simple and low-cost gene editing technologies available, and is a very popular gene editing system present.

SiRNA suppresses the expression of its target gene at the post-transcriptional level by mRNA degradation. Because of the capability of siRNA in selective targeting, much attention has been directed toward using siRNA in treatment of different cancers. However, this conventional method cannot remove the original copy of the oncogenes and the proteins are translated again with the next generation of genetic replication. Although both strategies of siRNA and CRISPR-Cas9 could down regulate gene editing, knockdown strategy based on CRISPR-Cas9 has the advantages of permanently silencing the target gene, high effectiveness for accurate gene editing, and low off-target likelihood. That's the reason we chose the CRISPR-Cas9 system.

3. Because PMZH is pH sensitive, was the dye released from PMZH and escaped from endo/lysosomes? The author needs a further explanation for this.

Our response: Thanks for your question. PMZH can effectively deliver pDNA to organelles through endocytosis, and further release it into the nucleus. This good endosome/lysosomal escape ability stems from the proton sponge effect of the imidazole ring. PMZH contains imidazole groups, which successfully introduces intracellular pH buffer element, thus improving transfection efficiency. The "proton sponge" hypothesis proposed a mechanism for this behavior, indicating that the buffer activity generates osmotic swelling and lysis of endocytic compartments. Reduced lysosomal degradation and greater cytoplasmic availability of delivered DNA would then give improvements in transfection efficiency.

4. The numerical changes in ROS detection in figure4a are not given, please supplement.

Our response: Thanks for your careful reading. We had supplemented its numerical changes (**Figure 5a**).

Figure 5. PMZH nanoplateform promotes STING signaling activation. **a** ROS generation in 4T1 cells assessed by flow cytometry after different treatments. **b** Mitochondrial membrane potential's changes in 4T1 cells after indicated treatments were evaluated by flow cytometry. **c** MtDNA copy numbers in 4T1 cells after different treatments were detected by real-time PCR. MtDNAs were normalized to β -actin mRNA encoded by the nuclear gene. **d** Immunoblot analysis of important proteins in the STING pathway in 4T1 cells after corresponding treatments. The expression of cGAS target genes **e** CXCL10 and **f** IFN- β in 4T1 cells was assessed by real-time PCR after different time periods of PMZH treatment. **g** Immunofluorescence staining of 4T1 cells with the indicated treatments. (Scale bar: 50 μ m). *P* values were assessed using Student's *t* test. Source data are provided as a Source Data file.

5. The authors claimed that modification with HA could enhance targeting effect, however, relevant data was not represented in the manuscript.

Our response: Thanks for your comments. We had complemented the relevant experiments in cells and in vivo. It can be seen that the coating of HA enhanced the recognition and accumulation at the tumor site, which due to the binding ability of HA to CD44 receptors on the surface of tumor cells (**Supplementary Figure 6, Supplementary Figure 17d**).

Supplementary Figure 6. Cellular uptake of FITC-modified PMZH and FITC-modified PMZ@PEG at 4 h.

Supplementary Figure 17. (a) Biodistribution of Zn in major organs and tumors (ID% per tissue Zn) after intravenous injection of PMZH at different time intervals (4, 24, and 72 h) (n = 3). (b) In vivo fluorescence images of mice after intravenous injection 24h of DiD-labeled PMZH. (c) ex-vivo fluorescence images of mice after intravenous injection 24h of Cy5-labeled PMZH. (d) Biodistribution of Zn in major organs and tumors (ID% per tissue Zn) after intravenous injection of PMZ@PEG and PMZH at 24h. Source data are provided as a Source Data file.

Reviewer #2 (Remarks to the Author): with expertise in cancer immunology, metabolism

Major concerns:

(1) In this study, the authors claimed that the nanoparticles could release ions in acid lysosome conditions and catalyze intracellular H₂O₂ to ROS to kill cancer cells. However, the tumor microenvironment (TME) is also acidic (pH 6.5-6.8) with a high concentration of H₂O₂. Are the nanoparticles stable in TME? Will they also catalyze TME H₂O₂ to ROS? Why does they affect T cell functions?

Our response: We sincerely appreciate the insightful comments of the reviewer. First, we evaluated the degradation of PMZH in a buffer mimicking the pH of the tumor microenvironment (pH = 6.5), and it can be seen that some degradation occurs in a buffer mimicking the TME (**Figure R1a**). Compared to the pH=6.5 group, PMZH undergoes faster and more complete degradation in the pH=5.5 group. The ability of PMZH to release pDNA in a pH-responsive manner is one of the most desirable properties for gene delivery and expressing systems (*J. Am. Chem. Soc.* 2018, 140, 1, 143–146; *Adv. Mater.* 2019, 31, 1901570.). Because of the tumor targeting ability endowed by HA (**Supplementary Figure 6**), most PMZH enters tumor cells in a short time rather than being degraded in the TME, it still has an effective plasmid delivery ability (**Figure 3a, Figure 4f-h**).

The Mn²⁺ from PMZH degradation could catalyze H₂O₂ into hydroxyl radicals in the TME (**Figure R1b**). Both intracellular and extracellular ROS produced by metal ions promotes STING pathway activation, thereby increasing T cell infiltration. As a result, the cytotoxic T cells (CD8⁺) and helper T cells (CD4⁺) expanded and infiltrated the neoplastic foci, which further reprogrammed the suppressive tumor microenvironment (TME) against the primary tumor and pulmonary metastases with safe systemic cytokine expression (*Adv. Sci.* 2023, 2300286. *Sci. Transl. Med.* 2022, 14, 648).

Figure R1. (a) Ion release at different time and under different conditions. The concentration of PMZH was 100 $\mu\text{g mL}^{-1}$. (b) UV-vis absorption spectra of MB after degradation by H₂O₂ (10 mM) plus different pH buffer-treated PMZH.

Supplementary Figure 6. Cellular uptake of FITC-modified PMZH and FITC-modified PMZ@PEG at 4 h.

Figure 3. Lyso/endosomal escape and gene editing. **a** Confocal microscopy images of PMZH co-incubated with 4T1 cells for 0.5, 1, 2 and 4 h to assess lyso/endosomal escape capacity. Plasmids were labeled with YOYO-1 (green). The scatter plot is the relationship between fluorescence signals in red and green. **b** Pearson's correlation coefficient (PCC) values of PMZH colocalized with lysosomes. The PCC values were calculated using ImageJ software. **c** Assessing the cellular uptake capacity of FITC-PMZH using flow cytometry. **d** Evaluation and **e** Semi-quantitative of MZH@EGFP transfection efficiency by fluorescence microscopy. **f** Evaluation of MZH@EGFP transfection efficiency by flow cytometry. **g** Sanger sequencing of PCR amplicons of target loci after PBS, CMZH, PMZH treatment. **h** WB analysis of *SLC43A2* expression after indicated treatments. (Scale bar: 50 μ m). Source data are provided as a Source Data file.

Figure 4. Effects of methionine metabolism regulation on T cell immunity. **a** Percentage of methionine consumption in cell culture medium after different treatments. Data were presented as mean \pm SD ($n = 3$ independent experiments). **b** Schematic representation of T cell co-culture with 4T1 cells in the Transwell system. **c** Flow cytometry to detect the changes of T cell viability in different treatment groups. **d** Flow cytometry quantitatively assesses the changes in the viability of 4T1 cells in different treatment groups. **e** Quantitative analysis of cytokines in the culture medium of different treatment groups. **f** Sanger sequencing of PCR amplicons of target loci after different treatment in tumor. **g** Western blot analysis of *SLC43A2* protein expression in tumors after indicated treatments. **h** Semi-quantitative analysis of *SLC43A2* expression in tumors. Effect of different treatments on apoptosis of **i** tumor cells and **j** tumor infiltrating CD8⁺ T cells in vivo. *P* values were assessed using Student's *t* test. Source data are provided as a Source Data file.

(2) Why the PMZH effectively knocked down SLC43A2 in tumors, but not in liver and kidney? The biodistribution of this nanosystem in organs should be performed.

Our response: Thanks for your question. We evaluated the biodistribution of PMZH and could see that PMZH was effectively enriched in the tumor (**Supplementary Figure 17c**). The HA modification enhanced the tumor recognition and accumulation by binding with CD44 receptors on tumor cells. HA modified nanomaterials could greatly reduce toxicity and side effects on non-tumor sites (*Adv. Mater.* 2019, 31, 1807211. *Nat. Commun.* 2022, 13, 6534). Because of the tumor targeting and the pH-responsive degradation ability of PMZH, it did not significantly affect SLC43A2 expression in the liver and kidney.

Supplementary Figure 17. (a) Biodistribution of Zn in major organs and tumors (ID% per tissue Zn) after intravenous injection of PMZH at different time intervals (4, 24, and 72 h) (n = 3). (b) In vivo fluorescence images of mice after intravenous injection 24h of DiD-labeled PMZH. (c) ex-vivo

fluorescence images of mice after intravenous injection 24h of Cy5-labeled PMZH. (d) Biodistribution of Zn in major organs and tumors (ID% per tissue Zn) after intravenous injection of PMZ@PEG and PMZH at 24h. Source data are provided as a Source Data file.

(3) The ROS increase upon nanoparticles accumulation in the tumor should be confirmed in vivo.

Our response: Thanks for your valuable comments. We had evaluated the production of ROS in tumors after different treatments, and PMZH caused ROS storm in tumors (Figure 6j).

Antitumor therapeutic effect of PMZH in vivo. **a** Schematic representation of vaccination and treatment. **b** Survival rates, **c** Body weights, **d** tumor volume changes, **e** TGI rates, **f** photographs of dissected tumors, **g** tumor weights of mice after different treatments. **h** MRI images after PMZH injection in 24 h. Data were presented as mean \pm SD ($n=4$ mice). P values were assessed using Student's t test. **i** H&E staining pictures of tumors after corresponding treatments. **j** ROS evaluation in tumors after different treatments. Source data are provided as a Source Data file.

(4) It has been known that Mn^{2+} dramatically promotes pro-inflammation response and anti-tumor immunity dependent on macrophages. Would the nanoparticles affect other immune cells (such as macrophages) and remodel the tumor immune microenvironment? How to demonstrate the nanoparticles specifically promotes CD8⁺ T cells antitumor function rather than the changes of the immune microenvironment?

Our response: Thanks for your valuable comments. Your insightful suggestion was very helpful for improving our study. We evaluated the effect of PMZH on macrophage polarization (**Figure R2**). PMZH could promote macrophage polarization to the anti-tumor M1 phenotype, means that PMZH could remodel the tumor immune microenvironment.

PMZH is not only promote CD8⁺T cells antitumor function, it had affected other immune cells. The immune system is a complex network, not separate parts, in addition, immune effector cells are interrelated. Our PMZH promoted macrophage polarization (**Figure R2**), activated the STING pathway of antigen-presenting cells (**Supplementary Figure 14**), and promoted the anti-tumor function of T cells. These synergistic effects make our immunotherapy more optimized.

We also supplemented experiments to further confirm the effect of PMZH on T cell immunity. PMZH restored T cell immunity through the regulation of methionine metabolism. Methionine deficiency decreased H3K79me2 in T cells (*Nature* 2020,585, 277–282), thereby impairing T cell immunity, and we assessed the effect of different treatments on H3K79me2 expression in T cells (**Supplementary Figure 10**). It can be seen that PMZH could restore H3K79me2 in T cells, which helps to restore T cell immunity.

Figure R2. Representative flow cytometric plots of macrophages (gated on F4/80+ cells) in tumor after various treatments.

Supplementary Figure 14. The secretion levels of (a) IFN- β and (b) CXCL10 in the supernatant of BMDCs after different treatments. Representative (c) flow cytometric plots and (d) quantitative analysis of mature BMDCs (CD80⁺CD86⁺ in CD11c⁺ cells) after incubation with various concentrations of PMZH. (e) Representative gating strategies for DC cells. Source data are provided

as a Source Data file.

Supplementary Figure 10. Western blot (a) shows H3K79me2 in CD8 T⁺ cells. (b) Semi-quantitative analysis of H3K79me2 expression. Source data are provided as a Source Data file.

(5) The authors claimed that PMZH remodels methionine metabolism in T cells. However, the authors don't provide any data to support that intervention of methionine metabolism in tumor cells could affect T cell immunity in this nanosystem.

Our response: Thanks for your valuable comments. We supplemented the effect of methionine concentration on the viability of tumor cells, T cells alone, and methionine concentration on the viability of co-cultured tumor cells and T cells (**Supplementary Figure 8**).

As shown in **Supplementary Figure 8a-b**, with the increase of methionine content, the apoptosis rate of 4T1 cells and T cells decreases, and T cells have a greater demand for methionine. When 4T1 cells and T cells were co-cultured in the Transwell system, with an increase in methionine concentration, 4T1 cell apoptosis was significantly alleviated, but T cells still had a higher apoptosis rate because 4T1 cells competed for methionine over T cells (**Supplementary Figure 8c-d**). Experimental results showed that tumor cells and T cells compete for methionine and that insufficient methionine uptake severely impairs T cell viability. This is consistent with previous research perspectives (*Nature* 2020,585, 277–282).

Supplementary Figure 8. Effect of methionine on viability of (a) 4T1 cells alone and (b) T cells alone. Effect of different concentrations of methionine on (c) 4T1 cells and (d) T cells viability when tumor and T cells compete for methionine in a Transwell system. Source data are provided as a Source Data file.

(6) lipo has cytotoxicity in cells. However, the authors showed that the lipo treatment in Figure 3C decreased the apoptosis of T cells. Please explain.

Our response: Thanks for your question. Firstly, lipo has cytotoxicity, and direct incubation with T cells using lipo can indeed lead to an increase in T cell apoptosis (Supplementary Figure 9).

However, the specific operation of **Figure 4c** (original **Figure3c**) is to treat 4T1 cells with different treatments first, and then co-culture the treated 4T1 cells with T cells. This

is to evaluate how the effects of different materials on tumor cells indirectly affect the vitality of T cells. During this process, Lipo does not come into direct contact with T cells.

In the **Figure 4c** (original **Figure3c**), co-cultured with 4T1 cells which was treated with Lipo decreased apoptosis of T cells. That's probably due to the apoptosis of tumor cells relieving T cell metabolic stress and thus decreasing apoptosis of T cells.

Supplementary Figure 9. In a non-Transwell system (a) Flow cytometry to detect the changes of 4T1 cell viability in different treatment groups. (b) Flow cytometry quantitatively assesses the changes in the viability of T cells in different treatment groups. Source data are provided as a Source Data file.

Figure 4. Effects of methionine metabolism regulation on T cell immunity. **a** Percentage of methionine consumption in cell culture medium after different treatments. Data were presented as mean \pm SD ($n = 3$ independent experiments). **b** Schematic representation of T cell co-culture with 4T1 cells in the Transwell system. **c** Flow cytometry to detect the changes of T cell viability in different treatment groups. **d** Flow cytometry quantitatively assesses the changes in the viability of 4T1 cells in different treatment groups. **e** Quantitative analysis of cytokines in the culture medium of different treatment groups. **f** Sanger sequencing of PCR amplicons of target loci after different treatment in tumor. **g** Western blot analysis of *SLC43A2* protein expression in tumors after indicated treatments. **h** Semi-quantitative analysis of *SLC43A2* expression in tumors. Effect of different treatments on apoptosis of **i** tumor cells and **j** tumor infiltrating CD8⁺ T cells in vivo. *P* values were assessed using Student's *t* test. Source data are provided as a Source Data file.

Minor concerns:

(1) The full name of ZIF should be spelled out in the manuscript.

Our response: Thanks for your careful comments. We had supplement the full names of ZIF (Mn/Zn-ZIF-8) in the manuscript.

(2) The quantitative colocalization analysis of plasmid with endo/lysosomes in 4T1 cells should be conducted. The color scatter plots and corresponding Pearson's correlation coefficient (PCC) values between the red and green fluorescence signals in the images should be provided.

Our response: Thanks to the reviewer for the professional comment. We had complemented quantitative colocalization analysis of plasmid with endo/lysosomes in 4T1 cells, and had complemented the color scatter plots and corresponding Pearson's correlation coefficient (PCC) values (**Figure 3a, b**).

Figure 3. Lyso/endosomal escape and gene editing. **a** Confocal microscopy images of PMZH co-incubated with 4T1 cells for 0.5, 1, 2 and 4 h to assess lyso/endosomal escape capacity. Plasmids were labeled with YOYO-1 (green). The scatter plot is the relationship between fluorescence signals in red and green. **b** Pearson's correlation coefficient (PCC) values of PMZH colocalized with lysosomes. The PCC values were calculated using ImageJ software. **c** Assessing the cellular uptake capacity of FITC-PMZH using flow cytometry. **d** Evaluation and **e** Semi-quantitative of MZH@EGFP transfection efficiency by fluorescence microscopy. **f** Evaluation of MZH@EGFP transfection efficiency by flow cytometry. **g** Sanger sequencing of PCR amplicons of target loci after PBS, CMZH, PMZH treatment. **h** WB analysis of *SLC43A2* expression after indicated treatments. (Scale bar: 50 μ m). Source data are provided as a Source Data file.

(3) The title of the y-axis in Figure 4A should be labeled.

Our response: Thanks for your comments. We had labeled the y-axis in **Figure 5a** (original Figure 4a).

Figure 5. PMZH nanoplateform promotes STING signaling activation. **a** ROS generation in 4T1 cells assessed by flow cytometry after different treatments. **b** Mitochondrial membrane potential's changes in 4T1 cells after indicated treatments were evaluated by flow cytometry. **c** MtDNA copy numbers in 4T1 cells after different treatments were detected by real-time PCR. MtDNAs were normalized to β -actin mRNA encoded by the nuclear gene. **d** Immunoblot analysis of important proteins in the STING pathway in 4T1 cells after corresponding treatments. The expression of cGAS target genes **e** CXCL10 and **f** IFN- β in 4T1 cells was assessed by real-time PCR after different time periods of PMZH treatment. **g** Immunofluorescence staining of 4T1 cells with the indicated treatments. (Scale bar: 50 μ m). *P* values were assessed using Student's *t* test. Source data are provided as a Source Data file.

(4) The quality of WB results should be improved (such as fig2f, fig6g).

Our response: Thanks for your comments. We had re-conducted the WB experiment to obtain higher quality images (**Figure 3h, Figure 4g, Figure 5d, and Figure 7g**).

Figure 3. Lyso/endosomal escape and gene editing. **a** Confocal microscopy images of PMZH co-incubated with 4T1 cells for 0.5, 1, 2 and 4 h to assess lyso/endosomal escape capacity. Plasmids were labeled with YOYO-1 (green). The scatter plot is the relationship between fluorescence signals in red and green. **b** Pearson's correlation coefficient (PCC) values of PMZH colocalized with lysosomes. The PCC values were calculated using ImageJ software. **c** Assessing the cellular uptake

capacity of FITC-PMZH using flow cytometry. **d** Evaluation and **e** Semi-quantitative of MZH@EGFP transfection efficiency by fluorescence microscopy. **f** Evaluation of MZH@EGFP transfection efficiency by flow cytometry. **g** Sanger sequencing of PCR amplicons of target loci after PBS, CMZH, PMZH treatment. **h** WB analysis of *SLC43A2* expression after indicated treatments. (Scale bar: 50 μ m). Source data are provided as a Source Data file.

Figure 4. Effects of methionine metabolism regulation on T cell immunity. a Percentage of methionine consumption in cell culture medium after different treatments. Data were presented as mean \pm SD ($n = 3$ independent experiments). **b** Schematic representation of T cell co-culture with 4T1 cells in the Transwell system. **c** Flow cytometry to detect the changes of T cell viability in different treatment groups. **d** Flow cytometry quantitatively assesses the changes in the viability of 4T1 cells in different treatment groups. **e** Quantitative analysis of cytokines in the culture medium of different treatment groups. **f** Sanger sequencing of PCR amplicons of target loci after different treatment in

tumor. **g** Western blot analysis of *SLC43A2* protein expression in tumors after indicated treatments. **h** Semi-quantitative analysis of *SLC43A2* expression in tumors. Effect of different treatments on apoptosis of **i** tumor cells and **j** tumor infiltrating CD8⁺ T cells in vivo. *P* values were assessed using Student's *t* test. Source data are provided as a Source Data file.

Figure 5. PMZH nanoplateform promotes STING signaling activation. **a** ROS generation in 4T1 cells assessed by flow cytometry after different treatments. **b** Mitochondrial membrane potential's changes in 4T1 cells after indicated treatments were evaluated by flow cytometry. **c** MtDNA copy numbers in 4T1 cells after different treatments were detected by real-time PCR. MtDNAs were normalized to β -actin mRNA encoded by the nuclear gene. **d** Immunoblot analysis of important proteins in the STING pathway in 4T1 cells after corresponding treatments. The expression of cGAS target genes **e** CXCL10 and **f** IFN- β in 4T1 cells was assessed by real-time PCR after different time periods of PMZH treatment. **g** Immunofluorescence staining of 4T1 cells with the indicated treatments. (Scale bar: 50 μ m). *P* values were assessed using Student's *t* test. Source data are provided as a Source Data file.

Figure 7. Immunity therapy of PMZH. a Flow cytometry analysis of DCs (gating on CD11c⁺) infiltrated after treatment with different formulations. Populations of b CD8⁺ and c CD4⁺T cells in 4T1 tumor tissues after different treatment. Cytokine expression levels of d TNF-α and e IL-6 in

serum were analyzed by Elisa kit after different treatments. **f** The IFN- β expression in tumors treated with different preparations was detected by qPCR. Data were presented as mean \pm SD ($n = 4$ mice). *P* values were assessed using Student's *t* test. **g** Immunoblotting analysis of STING pathway-related protein levels in tumor samples from various treatment groups. **h** Semi-quantitative analysis of p-TBK1 and IFN- β expression. Source data are provided as a Source Data file.

Reviewer #3 (Remarks to the Author): with expertise in cancer nano-therapy

1. The authors introduced a CMZH control group in their study in vivo to exclude the effect of a nonfunctional plasmid on methionine metabolism. However, they did not provide the effect of the CMZH control group on SLC43A2 gene expression. Please supplement. In addition, compared with CMZH, PMZH effectively reduced the level of apoptosis of T cells, but the effect of PMZH and CMZH on the secretion of functional cytokines was not evaluated. This needs further research and discussion.

Our response: Thanks for your professional comments. Your insightful suggestion was very helpful for improving our study. We had supplemented the related experiments on SLC43A2 gene expression by CMZH control group, and it could be seen that CMZH did not affect the expression of SLC43A2 gene (**Figure 3g-h, Figure 4f-h**). We further evaluated the effect of PMZH and CMZH on the secretion of functional cytokines (**Figure 4e**), and PMZH promoted the secretion of TNF- α more effectively than CMZH, which was consistent with our previous conclusion that PMZH effectively restored T cell immunity through methionine metabolism regulation.

Figure 3. Lyso/endosomal escape and gene editing. **a** Confocal microscopy images of PMZH co-incubated with 4T1 cells for 0.5, 1, 2 and 4 h to assess lyso/endosomal escape capacity. Plasmids were labeled with YOYO-1 (green). The scatter plot is the relationship between fluorescence signals in red and green. **b** Pearson's correlation coefficient (PCC) values of PMZH colocalized with lysosomes. The PCC values were calculated using ImageJ software. **c** Assessing the cellular uptake capacity of FITC-PMZH using flow cytometry. **d** Evaluation and **e** Semi-quantitative of MZH@EGFP transfection efficiency by fluorescence microscopy. **f** Evaluation of MZH@EGFP transfection efficiency by flow cytometry. **g** Sanger sequencing of PCR amplicons of target loci after PBS, CMZH, PMZH treatment. **h** WB analysis of *SLC43A2* expression after indicated treatments. (Scale bar: 50 μ m). Source data are provided as a Source Data file.

Figure 4. Effects of methionine metabolism regulation on T cell immunity. a Percentage of methionine consumption in cell culture medium after different treatments. Data were presented as mean \pm SD ($n = 3$ independent experiments). **b** Schematic representation of T cell co-culture with 4T1 cells in the Transwell system. **c** Flow cytometry to detect the changes of T cell viability in different treatment groups. **d** Flow cytometry quantitatively assesses the changes in the viability of 4T1 cells in different treatment groups. **e** Quantitative analysis of cytokines in the culture medium of different treatment groups. **f** Sanger sequencing of PCR amplicons of target loci after different treatment in tumor. **g** Western blot analysis of *SLC43A2* protein expression in tumors after indicated treatments. **h** Semi-quantitative analysis of *SLC43A2* expression in tumors. Effect of different treatments on apoptosis of **i** tumor cells and **j** tumor-infiltrating CD8⁺ T cells in vivo. *P* values were assessed using Student's *t* test. Source data are provided as a Source Data file.

2. Cytosolic DNA can be recognized by CGAs and other DNA sensors to produce type I interferons and antitumor innate immunity. Although the authors demonstrated the effects of methionine transport on the sting pathway in vivo, experiments at the cellular level are lacking. The CMZH group needs to be introduced in cytological experiments as well.

Our response: Thanks for your professional comments. We introduced a CMZH control group at the cellular level (**Figure 5**), and the experimental results were consistent with those in vivo (**Supplementary Figure 20**). Compared with CMZH, PMZH promoted ROS generation, mtDNA release, and STING signaling activation more efficiently. Methionine restriction can enhance the sensitivity of tumor cells to oxidative stress. This phenomenon indicated that the CRISPR plasmid regulated methionine metabolism, thereby enhancing ROS production and promoting the activation of the STING signaling pathway. This is consistent with the conclusion of the previous in vivo experiment.

Figure 5. PMZH nanoplateform promotes STING signaling activation. **a** ROS generation in 4T1 cells assessed by flow cytometry after different treatments. **b** Mitochondrial membrane potential's changes in 4T1 cells after indicated treatments were evaluated by flow cytometry. **c** MtDNA copy numbers in 4T1 cells after different treatments were detected by real-time PCR. MtDNAs were normalized to β -actin mRNA encoded by the nuclear gene. **d** Immunoblot analysis of important proteins in the STING pathway in 4T1 cells after corresponding treatments. The expression of cGAS target genes **e** CXCL10 and **f** IFN- β in 4T1 cells was assessed by real-time PCR after different time periods of PMZH treatment. **g** Immunofluorescence staining of 4T1 cells with the indicated treatments. (Scale bar: 50 μ m). *P* values were assessed using Student's *t* test. Source data are provided as a Source Data file.

Supplementary Figure 20. Western blot analysis of STING pathway-related protein levels in tumor samples from different treatment groups. Source data are provided as a Source Data file.

3. Please supplement the fluorescence intensity scale bar of biodistribution (supplement, Figure10b).

Our response: Thanks for your valuable comments. We had supplement the scale bar (**Supplementary Figure 17 b**).

Supplementary Figure 17. (a) Biodistribution of Zn in major organs and tumors (ID% per tissue Zn) after intravenous injection of PMZH at different time intervals (4, 24, and 72 h) (n = 3). (b) In vivo fluorescence images of mice after intravenous injection 24h of DiD-labeled PMZH. (c) ex-vivo fluorescence images of mice after intravenous injection 24h of Cy5-labeled PMZH. (d) Biodistribution of Zn in major organs and tumors (ID% per tissue Zn) after intravenous injection of PMZ@PEG and PMZH at 24h. Source data are provided as a Source Data file.

4. Flow gate strategies for DC cells, Figure7f-g should be provided in SI.

Our response: Thanks for your valuable comments. We had supplement the Flow gate strategies (**Supplementary Figure 12a-b, Supplementary Figure 14e**).

Supplementary Figure 12. Representative gating strategies for (a) CD45⁺ 4T1 cells apoptosis and (b) CD8⁺ T cells apoptosis.

Supplementary Figure 14. The secretion levels of (a) IFN- β and (b) CXCL10 in the supernatant of BMDCs after different treatments. Representative (c) flow cytometric plots and (d) quantitative analysis of mature BMDCs (CD80⁺CD86⁺ in CD11c⁺ cells) after incubation with various concentrations of PMZH. (e) Representative gating strategies for DC cells. Source data are provided as a Source Data file.

5. The targeting effect of HA needs to be characterized.

Our response: Thanks for your valuable comments. We had complemented the relevant experiments in cells (**Supplementary Figure 6**) and in vivo (**Supplementary Figure 17d**). It can be seen that the coating of HA enhanced the recognition and accumulation at the tumor site, which due to the binding ability of HA to CD44 receptors on the surface of tumor cells.

Supplementary Figure 6. Cellular uptake of FITC-modified PMZH and FITC-modified PMZ@PEG at 4 h.

Supplementary Figure 17. (a) Biodistribution of Zn in major organs and tumors (ID% per tissue Zn) after intravenous injection of PMZH at different time intervals (4, 24, and 72 h) (n = 3). (b) In vivo fluorescence images of mice after intravenous injection 24h of DiD-labeled PMZH. (c) ex-vivo fluorescence images of mice after intravenous injection 24h of Cy5-labeled PMZH. (d) Biodistribution of Zn in major organs and tumors (ID% per tissue Zn) after intravenous injection of PMZ@PEG and PMZH at 24h. Source data are provided as a Source Data file.

6. The authors should clarify why they choose plasmid DNA but not mRNA to encode the CRISPR/Cas9. Usually the immunogenicity of mRNA is much lower than plasmid DNA and can achieve higher protein expression in vivo.

Our response: Thank you very much for your valuable comments. Firstly, mRNA does not need to enter the nucleus for transcription. In addition, recent studies have also found that the addition of modified nucleotides can reduce the immunogenicity of

mRNA and improve the translation efficiency of proteins. These advantages all indicated that mRNA has priority in clinical applications.

In our study, we chose plasmid instead of mRNA, mainly considering the efficiency of protein self-supply. The CRISPR / cas9 system mainly consists of cas9 components and sgRNA components. The two components can be co constructed in a single plasmid system, but using mRNA requires two separate RNAs with a large difference in length (cas9 mRNA: ~ 5000 NT; sgRNA: ~ 300 NT). Determining the ratio of cas9 mRNA and sgRNA for optimal protein self-supply efficiency and regulating the efficiency of MOF for simultaneous loading of different lengths mRNA poses a challenge for the conduct of experiments. In addition, DNA is more stable than mRNA, which may make the protein self-supply of CRISPR/Cas9 more durable. Therefore, we chose CRISPR/Cas9 plasmid for this study.

7. Please check for formatting issues throughout this manuscript.

Our response: Thanks for your careful reading. We have tried our best to correct the formatting issues in the revised manuscript and re-examine the whole article to revise similar errors.

Reviewer #4 (Remarks to the Author): with expertise in cancer immunology, metabolism

My major comment is related to an unspecified mechanism of action of this nanoparticle. On the one hand, there is reduced tumor methionine consumption and higher T cell function. On the other hand, the enhanced T cell function could be due to a general enhancement in STING/cGAS activation or both. Some genetic loss of function experiments are essential to clarify how these nanoparticles are acting.

Our response: We sincerely appreciate the insightful comments of the reviewer. We performed further evaluations of the mechanism of PMZH affecting T cells. We assessed T cell viability and function after different treatments in wild-type 4T1 tumor-bearing mice and in STING-KO 4T1 tumor-bearing mice (**Supplementary Figure 23**). The results showed that the enhancement of T cell function by PMZH was more pronounced in wild-type 4T1 tumor-bearing mice. At the same time, we also supplemented the effects of CMZH and PMZH groups on T cells. We used pDNA inserting a non-targeting sgRNA as the control plasmid to construct CMZH. PMZH restored T cell immunity through the regulation of methionine metabolism. Methionine deficiency decreased H3K79me2 in T cells (*Nature* 2020,585, 277–282), thereby impairing T cell immunity, and we assessed the effect of different treatments on H3K79me2 expression in T cells (**Supplementary Figure 10**). It can be seen that PMZH could restore H3K79me2 in T cells, which helps to restore T cell immunity. These phenomena all indicated that the activation of the STING pathway increased T cell infiltration, while the regulation of methionine metabolism restored T cell function. The synergistic effect of the both effectively enhanced T cell immunity.

Supplementary Figure 23. (a) Effect of different treatments on apoptosis of tumor infiltrating CD8⁺ T cells in vivo. (b) Cytokine expression levels of TNF- α in serum were analyzed by Elisa kit after different treatments. *P* values were assessed using Student's *t* test (two-tailed) (**P*<0.01, ***P*<0.005, ****P*<0.001). Source data are provided as a Source Data file.

Supplementary Figure 10. Western blot (a) shows H3K79me2 in CD8⁺ T cells. (b) Semi-quantitative analysis of H3K79me2 expression. Source data are provided as a Source Data file.

A second comment is the use of PBS as a control in all the in vitro experiments. At a minimum there should be a non-targeting sgRNA as well as testing some individual components of the nano-particle itself. Ideally, the authors should do this on the tumor cells alone, T cells alone, and then both in a co-culture experiment.

Our response: Thanks to the reviewer for the professional comment. We used pDNA inserting a non-targeting sgRNA as the control plasmid to construct CMZH. Neither gene expression nor protein expression of *SLC43A2* was affected by CMZH (Figure 3g-h). In contrast to PMZH, CMZH does not have the ability to regulate methionine metabolism but the other components are all the same. We added the pDNA, Mn²⁺, Zn²⁺, and CMZH control groups in experiments in vitro as required (Figure 4a, c-e). As can be seen, in tumor cells alone and co-cultured tumor cells, pDNA alone did not affect cell viability; The Mn²⁺/Zn²⁺ mixture group and CMZH groups showed similar degrees of cell killing, while PMZH showed the strongest ability to kill tumor cells. This illustrated that the antitumor ability of PMZH not only stems from nutrient metal ions, but is also closely related to the gene editing effect of CRISPR. In the T cell alone group, there was no significant difference between CMZH and PMZH groups in T cell killing because T cells did not rely on SLC43A2 protein for methionine transport, unlike tumor cells. While in the co-culture group, PMZH limited tumor cell methionine uptake, relieved T cell metabolic stress, and thus reduced T cell apoptosis.

Figure 3. Lyso/endosomal escape and gene editing. **a** Confocal microscopy images of PMZH co-incubated with 4T1 cells for 0.5, 1, 2 and 4 h to assess lyso/endosomal escape capacity. Plasmids were labeled with YOYO-1 (green). The scatter plot is the relationship between fluorescence signals in red and green. **b** Pearson's correlation coefficient (PCC) values of PMZH colocalized with lysosomes. The PCC values were calculated using ImageJ software. **c** Assessing the cellular uptake capacity of FITC-PMZH using flow cytometry. **d** Evaluation and **e** Semi-quantitative of MZH@EGFP transfection efficiency by fluorescence microscopy. **f** Evaluation of MZH@EGFP transfection efficiency by flow cytometry. **g** Sanger sequencing of PCR amplicons of target loci after PBS, CMZH, PMZH treatment. **h** WB analysis of *SLC43A2*

expression after indicated treatments. (Scale bar: 50 μ m). Source data are provided as a Source Data file.

Figure 5. PMZH nanoplateform promotes STING signaling activation. **a** ROS generation in 4T1 cells assessed by flow cytometry after different treatments. **b** Mitochondrial membrane potential's changes in 4T1 cells after indicated treatments were evaluated by flow cytometry. **c** MtDNA copy numbers in 4T1 cells after different treatments were detected by real-time PCR. MtDNAs were normalized to β -actin mRNA encoded by the nuclear gene. **d** Immunoblot analysis of important proteins in the STING pathway in 4T1 cells after corresponding treatments. The expression of cGAS target genes **e** CXCL10 and **f** IFN- β in 4T1 cells was assessed by real-time PCR after different time periods of PMZH treatment. **g** Immunofluorescence staining of 4T1 cells with the indicated treatments. (Scale bar: 50 μ m). *P* values were assessed using Student's *t* test. Source data are provided as a Source Data file.

Third, most, if not all, of the micrographs need to be carefully quantified and

statistical analysis performed.

Our response: Thanks for your valuable comments. We had quantified and statistically analyzed fluorescence microscopy photographs (**Figure 3a-b, Figure 3d-e, Figure 4g-h, Figure 5g, Figure 7g-h, Supplementary Figure 10a-b, Supplementary Figure 20a-b**).

Figure 3. Lyso/endosomal escape and gene editing. **a** Confocal microscopy images of PMZH co-incubated with 4T1 cells for 0.5, 1, 2 and 4 h to assess lyso/endosomal escape capacity. Plasmids were labeled with YOYO-1 (green). The scatter plot is the relationship between

fluorescence signals in red and green. **b** Pearson's correlation coefficient (PCC) values of PMZH colocalized with lysosomes. The PCC values were calculated using ImageJ software. **c** Assessing the cellular uptake capacity of FITC-PMZH using flow cytometry. **d** Evaluation and **e** Semi-quantitative of MZH@EGFP transfection efficiency by fluorescence microscopy. **f** Evaluation of MZH@EGFP transfection efficiency by flow cytometry. **g** Sanger sequencing of PCR amplicons of target loci after PBS, CMZH, PMZH treatment. **h** WB analysis of *SLC43A2* expression after indicated treatments. (Scale bar: 50 μ m). Source data are provided as a Source Data file.

Figure 4. Effects of methionine metabolism regulation on T cell immunity. a Percentage of methionine consumption in cell culture medium after different treatments. Data were presented as mean \pm SD ($n = 3$ independent experiments). **b** Schematic representation of T cell co-culture with 4T1 cells in the Transwell system. **c** Flow cytometry to detect the changes of T cell viability in

different treatment groups. **d** Flow cytometry quantitatively assesses the changes in the viability of 4T1 cells in different treatment groups. **e** Quantitative analysis of cytokines in the culture medium of different treatment groups. **f** Sanger sequencing of PCR amplicons of target loci after different treatment in tumor. **g** Western blot analysis of *SLC43A2* protein expression in tumors after indicated treatments. **h** Semi-quantitative analysis of *SLC43A2* expression in tumors. Effect of different treatments on apoptosis of **i** tumor cells and **j** tumor infiltrating CD8⁺ T cells in vivo. *P* values were assessed using Student's *t* test. Source data are provided as a Source Data file.

Figure 5. PMZH nanoplateform promotes STING signaling activation. **a** ROS generation in 4T1 cells assessed by flow cytometry after different treatments. **b** Mitochondrial membrane potential's changes in 4T1 cells after indicated treatments were evaluated by flow cytometry. **c** MtDNA copy numbers in 4T1 cells after different treatments were detected by real-time PCR. MtDNAs were normalized to β -actin mRNA encoded by the nuclear gene. **d** Immunoblot analysis of important proteins in the STING pathway in 4T1 cells after corresponding treatments. The expression of cGAS target genes **e** CXCL10 and **f** IFN- β in 4T1 cells was assessed by real-time PCR after different time periods of PMZH treatment. **g** Immunofluorescence staining of 4T1 cells

with the indicated treatments. (Scale bar: 50 μm). *P* values were assessed using Student's *t* test.

Source data are provided as a Source Data file.

Figure 7. Immunity therapy of PMZH. **a** Flow cytometry analysis of DCs (gating on CD11c+) infiltrated after treatment with different formulations. Populations of **b** CD8⁺ and **c** CD4⁺T cells in 4T1 tumor tissues after different treatment. Cytokine expression levels of **d** TNF- α and **e** IL-6 in serum were analyzed by Elisa kit after different treatments. **f** The IFN- β expression in tumors

treated with different preparations was detected by qPCR. Data were presented as mean \pm SD ($n = 4$ mice). *P* values were assessed using Student's *t* test. **g** Immunoblotting analysis of STING pathway-related protein levels in tumor samples from various treatment groups. **h** Semi-quantitative analysis of p-TBK1 and IFN- β expression. Source data are provided as a Source Data file.

Supplementary Figure 10. Western blot (a) shows H3K79me2 in CD8⁺ T cells. (b) Semi-quantitative analysis of H3K79me2 expression. Source data are provided as a Source Data file.

Supplementary Figure 20. Western blot analysis of STING pathway-related protein levels in tumor samples from different treatment groups. Source data are provided as a Source Data file.

Forth, while the authors do examine DC markers and frequency of DCs, careful assessment of STING activation in these DC populations needs to be performed. In other words, does the nanoparticle activate STING in DCs or tumors or another

immune population (e.g. myeloid). The authors suggest this happens in DCs but do not show these data.

Our response: Thanks for your valuable comments. Your insightful suggestion was very helpful for improving our study. The activation of STING signaling in bone marrow-derived dendritic cells (BMDCs) was further evaluated by the method reported in the literature (*ACS Nano* 2023, 17, 5, 4495–4506). The results (**Supplementary Figure 14**) showed that PMZH could effectively enhance sting signaling in BMDCs, and promote BMDC cell maturation and secretion of IFN- β , thereby enhancing systemic antitumor immunity.

Supplementary Figure 14. The secretion levels of (a) IFN- β and (b) CXCL10 in the supernatant of BMDCs after different treatments. Representative (c) flow cytometric plots and (d) quantitative analysis of mature BMDCs ($CD80^+CD86^+$ in $CD11c^+$ cells) after incubation with various concentrations of PMZH. (e) Representative gating strategies for DC cells. Source data are provided as a Source Data file.

While I am fine with some methods being deposited in the supplemental information, this is not sufficient for the reader to understand how certain experiments were performed. Therefore, the key methods of the main figures need to be placed in the manuscript. In particular, there needs to more details on the replicates and statistical test for each figure panel embedded directly in the captions.

Our response: Thanks for your valuable comments. We have supplemented important experimental methods of the figures in the **Methods** section of the manuscript. In addition, all bar graphs had replaced with plots that feature information about the distribution of the underlying data, and the details of replication and statistical testing had supplemented in the figure legend.

REVIEWERS' COMMENTS

Reviewer #1 (Remarks to the Author):

The authors have addressed my concerns. The revised manuscript is recommended to be accepted for Nature Communications.

Reviewer #2 (Remarks to the Author):

With the comments being addressed now, I endorse the publication.

Reviewer #3 (Remarks to the Author):

The revised manuscript adds substantial new data further confirming key points of the original submission. I appreciate the effort made by the authors to thoroughly respond to the reviewer comments. I find the revised manuscript substantially improved. I suggest this manuscript to be published without further revision.

Reviewer #4 (Remarks to the Author):

I have reviewed the revised manuscript. The authors have adequately addressed all of my concerns.

Reviewer #1 (Remarks to the Author):

The authors have addressed my concerns. The revised manuscript is recommended to be accepted for Nature Communications.

Response: Thanks for the reviewer's recognition and support of our work.

Reviewer #2 (Remarks to the Author):

With the comments being addressed now, I endorse the publication.

Response: Many thanks for the comments and suggestions of the reviewer, which are very helpful for improving the quality of the manuscript.

Reviewer #3 (Remarks to the Author):

The revised manuscript adds substantial new data further confirming key points of the original submission. I appreciate the effort made by the authors to thoroughly respond to the reviewer comments. I find the revised manuscript substantially improved. I suggest this manuscript to be published without further revision.

Response: We thank the reviewer for his/her comment and recommendation to publish.

Reviewer #4 (Remarks to the Author):

I have reviewed the revised manuscript. The authors have adequately addressed all of my concerns.

Response: We are truly grateful to your valuable comments and approval.